# EngiAgent: Fully Connected Coordination of LLM Agents for Solving Open-ended Engineering Problems with Feasible Solutions

## Abstract

Engineering problem solving is central to real-world decision-making, requiring mathematical formulations that not only represent complex problems but also produce feasible solutions under data and physical constraints. Unlike mathematical problem solving, which operates on predefined formulations, engineering tasks demand open-ended analysis, feasibility-driven modeling, and iterative refinement. Although large language models (LLMs) have shown strong capabilities in reasoning and code generation, they often fail to ensure feasibility, which limits their applicability to engineering problem solving. To address this challenge, we propose EngiAgent, a multi-agent system with a fully connected coordinator that simulates expert workflows through specialized agents for problem analysis, modeling, verification, solving, and solution evaluation. The fully connected coordinator enables flexible feedback routing, overcoming the rigidity of prior pipeline-based reflection methods and ensuring feasibility at every stage of the process. This design not only improves robustness to diverse failure cases such as data extraction errors, constraint inconsistencies, and solver failures, but also enhances the overall quality of problem solving. Empirical results across four representative domains demonstrate that EngiAgent achieves substantial improvements in feasibility compared to prior approaches, establishing a new paradigm for feasibility-oriented engineering problem solving with LLMs. Our source code and data are available at https://anonymous.4open.science/r/EngiAgent-1C8A.

## 1 Introduction

Large language models (LLMs) have shown strong capabilities in mathematical reasoning (Cobbe et al., 2021; Ahmaditeshnizi et al., 2024; Huang et al., 2025) and code generation (Wang et al., 2024; Dong et al., 2024; Wang et al., 2025), which has raised expectations that they could also solve real-world engineering problems. However, engineering problems, such as designing transportation schedules or coordinating autonomous systems, are different from mathematical or coding tasks. They require not only correct formulations but also feasible solutions that must work under physical, operational, and safety constraints. In engineering, feasibility takes priority. Yet this requirement is largely ignored in current LLM research, creating a gap between recent progress and the practical demands of engineering problem solving.

Although prior work has explored diverse areas, feasibility has rarely been prioritized, limiting applicability to engineering problems. Research on autonomous research has considered open-ended exploration (Yamada et al., 2025; Schmidgall et al., 2025; Lu et al., 2024), but the focus is usually on generating novel ideas rather than producing executable and logically consistent solutions. As a result, our empirical analysis shows that fewer than 10% of the solutions are feasible. Research in mathematical modeling often stresses formulation quality over feasibility (Liu et al., 2025b; Astorga et al., 2025), with limited attention to generating feasible numerical solutions. A state-of-the-art (SOTA) model has been reported to achieve nearly 70% success on mathematical benchmarks, yet in our evaluation it generates numerical solutions for only about 13% of engineering problems. Research on code generation are effective in generating executable programs (Guo et al., 2024; Gao et al., 2023) but often neglects structured data requirements and physical constraints. Our empirical results show that while 62.26% of problems generate numerical outputs, only 5.66% are feasible. Overall,

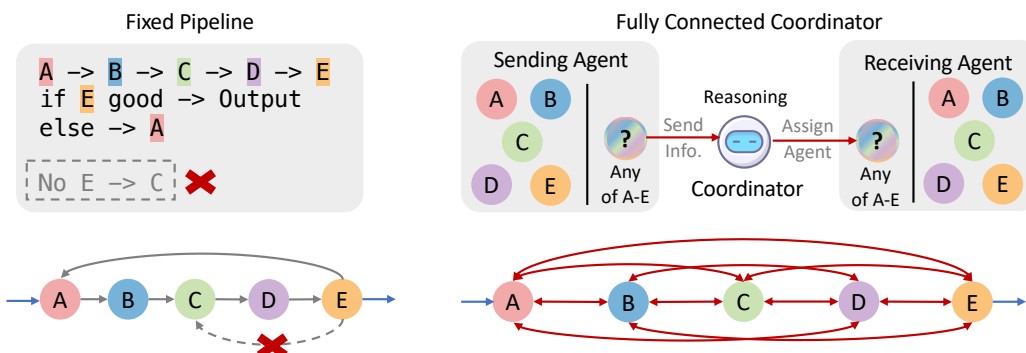

Figure 1: Comparison of fixed pipeline and fully connected coordinator. Fixed pipelines enforce rigid execution, whereas the fully connected coordinator enables dynamic routing and flexible error handling across agents.

prior work has advanced open-ended exploration, mathematical formulation, and code generation, but consistently neglects the core requirement of producing feasible solutions in engineering.

Ensuring feasibility is challenging since engineering problems require solutions based on given data and constrained by physical and operational realities. Beyond producing correct formulations or executable code, solutions also need to satisfy physical laws, safety requirements, operational rules, and domain-specific data conditions. Feasibility may break down at multiple stages, including information extraction, constraint specification, and solver execution, where even minor errors can render solutions infeasible. In practice, solving complex problems often relies on agent-based systems. However, existing designs, whether fixed pipelines or single-agent setups, lack the flexibility to coordinate across stages and correct diverse sources of error.

To overcome these limitations, we propose EngiAgent, a framework that integrates multi-agent systems with a fully connected coordinator. The framework simulates expert practice by dividing the workflow into dedicated roles for problem analysis, modeling, verification, solving, and solution evaluation. These agents collaborate to ensure feasibility at every stage while maintaining modeling quality. Unlike prior methods that rely on a fixed pipeline, the fully connected coordinator (Figure 1) dynamically routes feedback among agents, enabling targeted corrections when feasibility issues arise in data processing, constraint handling, or solver execution. This flexible structure enhances robustness and supports generalization across diverse engineering scenarios.

Empirical results demonstrate that EngiAgent achieves SOTA performance across three leading LLMs. The feasibility rates reach 64.15% on GPT-4o, 50.94% on Gemini-2.5 Flash, and 75.47% on DeepSeek-V3-671B, representing an average seven-fold improvement over prior SOTA methods. Furthermore, our fully connected coordinator improves feasibility by more than 10% on average compared with the fixed-pipeline framework.

Our contributions can be summarized as follows: (1) Feasibility is identified as a critical requirement for engineering modeling, and we collect a dataset covering four engineering domains with 53 high-quality problems to evaluate this aspect. (2) EngiAgent is proposed as the first framework designed to solve open-ended engineering problems by producing feasible solutions through a multi-agent system with a fully connected coordinator. (3) Empirical results demonstrate significant improvements in feasibility compared with previous approaches across diverse engineering tasks.

## 2 RELATED WORKS

**LLM Agents for Complex Problem.** With the advancement of LLMs, agents that combine autonomous reasoning, multi-step planning, and tool interaction have emerged as a new paradigm for complex problem solving (Huang et al., 2024b). Existing approaches, however, are typically built on fixed pipelines with predetermined workflows. They can be grouped into three categories: sequential planning agents that execute step-by-step processes (Yao et al., 2023; Yang et al., 2023), hierarchical task decomposition methods (Shen et al., 2023; Liang et al., 2024), and multi-agent coordination frameworks with fixed roles and protocols (Chen et al., 2023; Pan et al., 2025; Grötschla et al., 2025). These systems rely on static routing and lack adaptive error handling across formulation, modeling, verification, and solving.

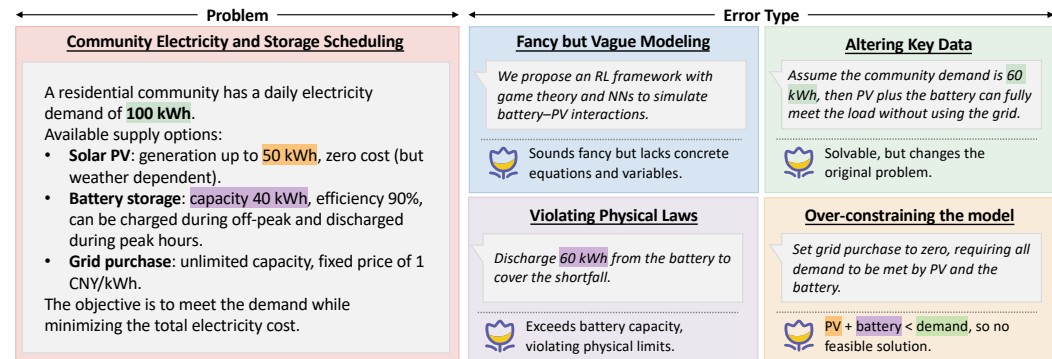

Figure 2: Illustration of common error types in engineering problem solving using a community electricity and storage scheduling problem. The four representative categories include fancy but vague modeling, altering key data, violating physical laws, and over-constraining the model.

Such rigidity becomes a bottleneck in open-ended engineering problems, which feature multi-objective trade-offs, evolving requirements, and interdependent constraints. Recent studies report systematic failure modes in multi-agent systems, including poor design, mis-specified tasks, inter-agent misalignment, and verification deficiencies (Cemri et al., 2025). Unlike structured computational tasks with clear success metrics, engineering challenges demand adaptive formulation, dynamic constraint handling, and trade-off navigation under real-world feasibility requirements (Mushtaq et al., 2025). This gap underscores the need for flexible coordination mechanisms that can dynamically adapt, debug, and refine workflows for engineering problem solving.

**LLMs for Engineering.** In recent years, advances in LLM reasoning have expanded their applications in engineering. With cross-domain knowledge, multi-step reasoning, and planning abilities, LLMs provide new pathways for modeling and solving complex engineering tasks. However, most studies remain confined to closed tasks and specific scenarios, lacking general modeling and iterative solving capabilities for open-ended engineering problems. Existing work has largely focused on power systems (Yan & Xu, 2023; Cheng et al., 2025; Majumder et al., 2024), materials (Liu et al., 2024; Jiang et al., 2025; Buehler, 2024), structures (Antunes et al., 2024; Jiang et al., 2023), mechanical design (Buehler, 2024; Xiaorui et al., 2025; Ni & Buehler, 2024), manufacturing (Zhou et al., 2024; Xia et al., 2024), and transportation (Yang et al., 2024; Qu et al., 2023), typically using fixed templates or tool-calling pipelines. While effective in predefined contexts, these approaches struggle with open problems involving uncertain objectives, complex constraints, and dynamic feedback, and fall short of building general systems for cross-task modeling, solving, and optimization.

Related research is autonomous research, which seeks to automate the scientific workflow from problem generation and literature review to experiment execution and paper writing. Current studies fall into two directions: multi-stage language generation and planning for general scientific tasks (Yamada et al., 2025; Schmidgall et al., 2025; Lu et al., 2024; Huang et al., 2024a; Liu et al., 2025a; Baek et al., 2025), and domain-specific automation in areas such as biomedical discovery (Gao et al., 2024), chemistry (Darvish et al., 2025), and transportation analysis (Guo et al., 2025). Compared with autonomous research, which emphasizes generating scientifically meaningful ideas and problem formulations, open-ended engineering automation focuses on structuring real-world problems, integrating professional tools and physical constraints, and producing feasible solutions through iterative feedback. This requires structured modeling, optimized tool use, feasibility verification, and adaptive refinement, extending autonomous research toward practical engineering problem solving.

## 3 FEASIBILITY CHALLENGES IN ENGINEERING PROBLEM SOLVING

In engineering problems, feasibility must take priority. The goal is not only to derive a formally correct model or numerical result, but to obtain a solution that can be implemented under real-world conditions. A solution that is logically consistent and mathematically valid becomes meaningless if it violates data consistency, physical laws, operational safety, or procedural constraints. Thus, the value of an engineering solution depends primarily on its feasibility.

In this work, a solution is considered feasible only if it satisfies all mandatory engineering constraints required for execution in real systems, such as data consistency, physical requirements, safety limits, and procedural rules.

This stands in contrast to mathematical problems, where correctness directly implies feasibility. Engineering problems are inherently open-ended, allowing multiple modeling choices. For example, simple versus complex formulations or linear versus nonlinear optimization. While many alternatives may appear logically sound, only a small fraction yield solutions that remain feasible under real-world constraints. This difference makes feasibility particularly difficult to ensure in engineering problem solving, since logical correctness alone does not guarantee implementability.

As illustrated in Figure 2, the common types of errors in engineering problem solving can be summarized into four categories:

1. Fancy but Vague Modeling: Methods that employ sophisticated terminology or advanced paradigms but fail to specify explicit variables, constraints, or equations. Such formulations sound appealing but cannot be transformed into computable or executable solutions.

2. Altering Key Data: Approaches that arbitrarily modify or replace critical data or conditions from the original problem. While this may lead to solvable models, the results no longer correspond to the intended problem and thus lack practical validity.

3. Violating Physical Laws: Models that ignore fundamental physical principles or engineering constraints, producing solutions that may be mathematically valid but physically impossible, such as exceeding capacity, efficiency, or safety limits.

4. Over-constraining the Model: Solutions that introduce excessive or unrealistic constraints. Although the formulation may appear more rigorous, the additional restrictions can eliminate all feasible solutions under real-world conditions.

Thus, although existing methods have made progress in areas such as open-ended exploration, mathematical modeling, and code generation, they still struggle to ensure feasibility in engineering contexts. A key reason is that these approaches often sacrifice feasibility in pursuit of other objectives. For instance, open-ended exploration emphasizes novelty, which frequently results in *fancy but vague modeling*. Mathematical modeling tends to focus on proposing elegant frameworks rather than executable solutions, which can lead both to *vague formulations* and to *over-constraining the model*. Code generation methods, while capable of producing runnable programs, often achieve this by altering data or modifying constraints, thereby introducing errors such as *altering key data* or *violating physical laws*.

These observations highlight a limitation of current research: they advance correctness and expressiveness but fail to guarantee feasibility, which is the essence of engineering problem solving. Addressing this gap requires new frameworks that explicitly prioritize feasibility at every stage of the modeling and solving process, ensuring that solutions are not only logically valid but also applicable in real-world practice.

## 4 ENGIAGENT

**Overview.** EngiAgent is a multi-agent system for solving open-ended engineering problems with feasible solutions. It consists of five functional agents: Analyzer, Modeler, Verifier, Solver, and Evaluator. Each agent is responsible for a specific stage, including problem analysis, code generation, model verification, code solving, and solution evaluation, which reflects the standard workflow commonly used in engineering problem solving (Hillier, 2005). Prior frameworks typically adopt fixed pipelines, which limits adaptability. In contrast, EngiAgent introduces a Fully Connected Coordinator that flexibly directs agent interactions and leverages Memory to support error diagnosis, adaptive scheduling, and cross-agent cooperation. This design moves beyond rigid pipelines and enables robust and efficient coordination across the end-to-end workflow (Figure 3).

**Fully Connected Coordinator.** The Fully Connected Coordinator serves as the control center, which manages the entire automated modeling and solving workflow. As shown in Figure 1, the coordinator integrates responses from specialized agents, applies adaptive reasoning as needed, and dispatches tasks to the corresponding agents for further processing. Under its coordination, each stage can produce results in line with the expectations of the agent role, and lay the foundation for

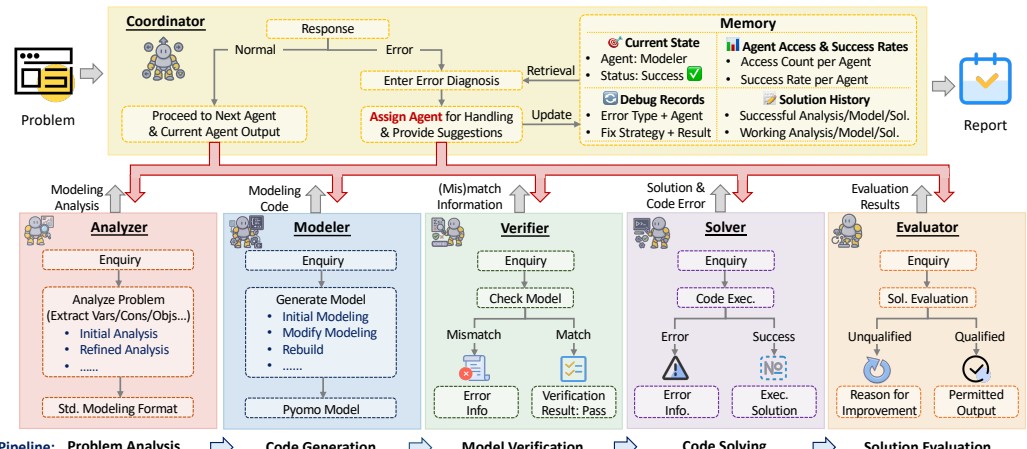

Figure 3: Overall Workflow of EngiAgent. The system consists of five functional agents (Analyzer, Modeler, Verifier, Solver, Evaluator) together with a Fully Connected Coordinator and shared Memory. The baseline pipeline (blue arrows) shows the predefined sequential cognitive architecture, while the coordinator layer (red arrows) enables flexible routing, error diagnosis, and adaptive scheduling across agents, ensuring feasibility and efficiency in solving open-ended engineering problems.

the subsequent stages. For example, the modeling analysis generates an accurate guidance based on the original problem, and the modeling under this guidance is compatible with the environment.

The coordinator connects the functions of all the agents to achieve an organic and coordinated operation. The rule-based scheduling within the coordinator controls the main flow from the Analyzer to the Evaluator. Its routing decisions are determined based on regular responses, verification results, and restart conditions. Furthermore, the coordinator can also make flexible decisions by leveraging LLM-based diagnosis. This module employs LLM-based decision making by providing agent success rates, debugging records and error details to analyze error types, directing constraint issues to the Analyzer and syntax problems to the Modeler. It also implements forced agent switching mechanism to prevent debugging loops from repeated similar errors. Additionally, to ensure the practicality and authenticity of final solutions, the coordinator appropriately considers Evaluator quality scores to make reliable continuation or termination decisions, maintaining detailed adaptation reports for all solution packages throughout the complete process.

**Analyzer.** The Analyzer serves as the starting point of the modeling process, converting natural language descriptions of engineering problems into structured problem formulations through LLM-based semantic parsing. It processes raw problem statements alongside knowledge of library retrieval to extract decision variables, parameters, constraints, and objectives. In addition to extracting explicit information, the Analyzer identifies implicit engineering rules and physical constraints that may not be explicitly expressed in the text, and further addresses uncertainty, trade-offs and sensitivity analysis. Dealing with retry requests, the agent incorporates mismatch information from verification and targeted modification strategies under coordinator guidance, and it supports reanalysis in response to code errors by adapting the modeling approach. The output is a standardized, hierarchical JSON format including modeling context, core model elements, and extended analysis, serving as a reliable foundation for subsequent stages.

**Modeler.** The Modeler functions as the core modeling engine, automatically transforming structured problem formulations into executable Pyomo optimization models through template-driven code generation. This agent integrates intelligent Pyomo-specific validation to prevent common errors such as inconsistent indexing, improper constraint formatting, or incorrect variable definitions, with special attention to model compatibility. The Modeler conducts modeling and leverages domain modeling experience to mitigate pitfalls frequently encountered in engineering contexts, ensuring the accurate use of JSON-based specifications, relational data input, and variable invocation. When retry requests are triggered, the system applies targeted corrections based on coordinator-provided strategies, focusing on Pyomo expression issues and proper adaptation between problem types and solution formats. It also contains optimization mechanisms for efficient handling of large-scale problems while preserving mathematical rigor and computational feasibility.

**Verifier.** The Verifier provides a quality assurance layer to ensure that the generated optimization models remain faithful to original engineering problems. It performs multi-dimensional verification covering semantic alignment, constraint completeness, variable validity, objective correctness, and data consistency. This agent examines correspondence between textual problem descriptions and formal mathematical formulations, to guarantee the complete utilization of background information and the coherent realization of model construction. The Verifier outputs detailed mismatch information with specific reasons and refinement recommendations concerning detected inconsistencies, which are subsequently returned to the Analyzer by coordinator for adjustment. It distinguishes between critical logical errors and acceptable simplifications, focusing on directional correctness and core requirement satisfaction. The Verifier either approves models for solver execution or initiates targeted refinement cycles, thereby reducing potential downstream issues that could compromise solution quality or computational efficiency.

**Solver.** The Solver executes optimization tasks with solver management and performance monitoring in isolated execution environments with enhanced robustness. The agent identifies solvers suitable for each problem, encompassing open-source packages such as GLPK and CBC as well as commercial tools like Gurobi and CPLEX, and employs a thread-based timeout strategy tailored to solver characteristics, to prevent deadlocks and excessive resource utilization. Moreover, this agent generates adequate feedback distinguishing between successful results and code errors, with detailed output information to support targeted debugging. Consequently, the Solver contributes to reliable optimization while maintaining stability and resource efficiency.

**Evaluator.** The Evaluator conducts comprehensive quality assessment of complete engineering optimization solutions and offers opportunities for improvement, balancing technical rigor with practical feasibility. It evaluates solution packages containing problem descriptions, modeling analyses, implementation code, and computational results across four key dimensions: result feasibility, model-problem alignment, engineering validity, and overall solution quality. This agent performs multi-perspective analysis and quality scoring utilizing LLM-driven assessment to provide constructive recommendations. The Evaluator generates detailed reports with scores, and makes restart or output decisions based on the overall evaluation. Through this holistic verification process, it ensures that final solutions meet practical standards while identifying targeted areas for optimization.

## 5 CONSTRUCTION OF THE FEASIBLE SOLUTION VALIDATION

**Problem Collection.** We construct a benchmark to explicitly evaluate the feasibility of solutions in engineering problem solving. The benchmark covers four domains: (1) market and multi-agent decision-making, (2) scheduling and resource allocation, (3) planning and design, and (4) control and autonomous system modeling. Problems are collected from high-quality peer-reviewed papers and rewritten into self-contained open-ended modeling tasks with background context, numerical data, and explicit instructions. Each task is further equipped with explicit engineering constraints derived from both the original papers and practical real-world requirements. These constraints define the conditions that must be satisfied for a solution to be considered feasible under realistic scenarios. In addition, following common practice in studies of modeling quality evaluation, we provide four dimensions to assess modeling and reasoning quality: information extraction, domain-specific reasoning, multi-objective decision-making, and uncertainty handling (Zhou et al., 2025; Liu et al., 2025b). To guarantee quality and relevance, more than 20 experienced domain experts reviewed and refined all tasks. Details of the dataset are provided in the Appendix.

**Task Formation.** Each task requires the model to extract key information from the text, including variables, parameters, constraints, and objectives, and then construct a numerical model to produce concrete solutions. The tasks are designed to preserve openness while still guiding the process toward verifiable numerical solving, which enables a faithful evaluation of the modeling and reasoning capabilities of LLMs in engineering problems.

**Evaluation.** Model performance is first evaluated by feasibility, which is the most critical dimension. Feasibility determines whether the final solution is consistent with the core data provided in the problem statement and whether it satisfies all specified constraints. It is assessed as a binary variable, where 0 indicates infeasible and 1 indicates feasible, based on executable numerical and constraint verification with final confirmation by human experts rather than any LLM-based scoring. Beyond feasibility, the quality of modeling and reasoning is evaluated along four dimensions, quantitatively scored on a scale from 0 to 10.

Table 1: Experimental results on open-ended engineering solving. Total (all problems) and Feasible (subset of feasible solutions) are reported. The Feas. column (highlighted in gray) is the key metric, representing the essential ability to generate solutions that satisfy real-world constraints. EngiAgent (Coord.) is our proposed system with a fully connected coordinator. Best results are in red, second best in blue. Abbreviations: Num. = proportion of problems that yield numerical solutions, Feas. = proportion of problems that yield feasible solutions, IE = Information Extraction, DR = Domain-specific Reasoning, MO = Multi-objective Decision-making, UH = Uncertainty Handling, Avg. = Average score.

| Methods | Num. ↑ | Feas. ↑ | Total | | | | | Feasible | | | | |
|---|---|---|---|---|---|---|---|---|---|---|---|---|
| | | | IE ↑ | DR ↑ | MO ↑ | UH ↑ | Avg. ↑ | IE ↑ | DR ↑ | MO ↑ | UH ↑ | Avg. ↑ |
| **GPT-4o** | | | | | | | | | | | | |
| Zero-shot | 22.64% | 5.66% | 5.66 | 5.42 | 4.47 | 3.33 | 4.72 | 5.67 | 5.10 | 1.33 | 4.00 | 4.03 |
| ResearchAgent | 15.09% | 3.77% | 5.36 | 5.25 | 5.10 | 5.17 | 5.22 | 5.50 | 5.50 | 6.00 | 5.50 | 5.63 |
| DS-Agent | 62.26% | 5.66% | 5.91 | 4.83 | 4.85 | 4.79 | 5.10 | 6.00 | 5.75 | 5.38 | 6.25 | 5.85 |
| MM-Agent | 13.21% | 7.55% | 6.89 | 7.21 | 6.48 | 7.98 | 7.14 | 8.25 | 8.38 | 7.88 | 7.50 | 8.00 |
| EngiAgent (Fixed) | 47.17% | 47.17% | 8.30 | 7.22 | 6.67 | 7.14 | 7.33 | 8.84 | 8.12 | 7.40 | 7.56 | 7.98 |
| EngiAgent (Coord.) | 66.04% | 64.15% | 8.67 | 7.74 | 7.05 | 7.41 | 7.72 | 8.86 | 8.13 | 7.53 | 7.94 | 8.12 |
| **Gemini-2.5 Flash** | | | | | | | | | | | | |
| Zero-shot | 0.00% | 0.00% | 6.80 | 5.67 | 5.53 | 3.55 | 5.39 | / | / | / | / | / |
| ResearchAgent | 1.89% | 0.00% | 4.56 | 4.41 | 4.34 | 4.36 | 4.42 | / | / | / | / | / |
| DS-Agent | 60.38% | 0.00% | 7.17 | 6.75 | 6.71 | 5.85 | 6.62 | / | / | / | / | / |
| MM-Agent | 9.43% | 3.77% | 6.39 | 5.72 | 5.98 | 6.40 | 6.12 | 8.00 | 8.00 | 8.25 | 6.00 | 7.56 |
| EngiAgent (Fixed) | 45.28% | 39.62% | 7.30 | 6.93 | 5.82 | 5.21 | 6.32 | 8.60 | 6.71 | 6.14 | 5.36 | 6.70 |
| EngiAgent (Coord.) | 52.83% | 50.94% | 8.30 | 6.89 | 6.30 | 6.06 | 6.89 | 8.80 | 7.23 | 6.50 | 6.81 | 7.34 |
| **DeepSeek-V3-671B** | | | | | | | | | | | | |
| Zero-shot | 0.00% | 0.00% | 6.20 | 5.60 | 5.09 | 3.99 | 5.22 | 6.50 | 6.25 | 6.50 | 6.50 | 6.44 |
| ResearchAgent | 9.43% | 7.55% | 5.12 | 4.89 | 4.84 | 4.98 | 4.96 | 6.50 | 5.98 | 4.92 | 4.98 | 5.60 |
| DS-Agent | 77.36% | 28.30% | 7.39 | 6.88 | 6.77 | 6.01 | 6.76 | 8.22 | 7.64 | 7.38 | 7.12 | 7.59 |
| MM-Agent | 11.32% | 5.66% | 8.98 | 8.84 | 7.64 | 7.01 | 8.12 | 9.00 | 8.50 | 8.00 | 7.85 | 8.34 |
| EngiAgent (Fixed) | 73.58% | 67.92% | 8.08 | 7.28 | 7.20 | 6.98 | 7.39 | 8.38 | 7.45 | 7.53 | 7.33 | 7.67 |
| EngiAgent (Coord.) | 79.25% | 75.47% | 7.85 | 7.42 | 7.08 | 6.59 | 7.24 | 8.05 | 8.16 | 7.26 | 6.52 | 7.50 |

# 6 EXPERIMENTS

## 6.1 BASELINES.

We evaluate EngiAgent against SOTA LLM agents for problem solving. Since no prior work explicitly targets open-ended engineering problem solving with feasibility requirements, we adapt existing LLM-based agents for comparison. The baselines include: (1) Zero-shot prompting, where the LLM directly attempts to solve problems without structured guidance; (2) ResearchAgent (Huang et al., 2024a), originally designed to automate experimentation loops in machine learning, which we extend to engineering tasks; (3) DS-Agent (Guo et al., 2024), an agent framework for automated data science based on case-based reasoning, adapted here to engineering modeling and solving; and (4) MM-Agent (Liu et al., 2025b), a framework for mathematical modeling tasks, repurposed for engineering problems by evaluating its ability to construct models and generate solutions.

## 6.2 EXPERIMENTAL RESULTS

**Limitations of existing methods.** Experiments (Table 1) show that existing frameworks achieve very low feasibility rates on engineering tasks. Each method has partial strengths: DS-Agent attains a relatively high rate of producing numerical solutions, but most outputs either mismatch the problem data or violate constraints, making them infeasible. MM-Agent can generate more feasible solutions, yet its overall ability to produce numerical results is too limited, often remaining at the level of textual modeling. Concretely, the best baseline feasible rates are only 7.55% on GPT-4o, 3.77% on Gemini-2.5 Flash, and 28.30% on DeepSeek-V3-671B.

**Overall advantage of EngiAgent.** On all three tested LLMs, EngiAgent (Coord.) consistently surpasses the strongest prior baselines in generating feasible solutions. On GPT-4o, the feasibility rate increases from 7.55% for the best baseline to 64.15% with EngiAgent. On Gemini-2.5 Flash, it rises from 3.77% to 50.94%. On DeepSeek-V3-671B, it increases from 28.30% to 75.47%. These results clearly highlight EngiAgent's substantial advantage in delivering feasible engineering solutions.

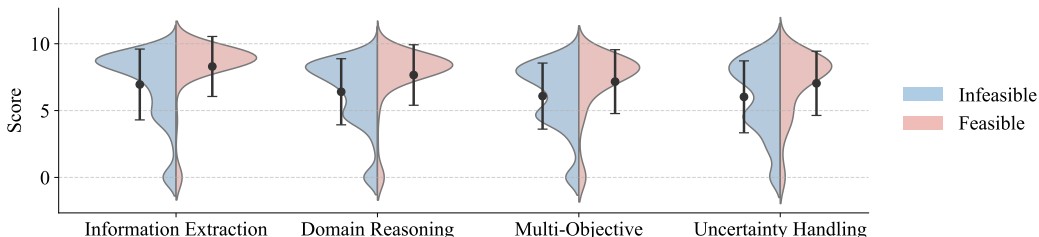

Figure 4: Distribution of evaluation scores across four dimensions for feasible and infeasible solutions. The actual maximum score is 10; values beyond 10 arise only from kernel density estimation (KDE) smoothing in the violin plot and do not occur in the underlying data.

**EngiAgent reduces the gap between numerical and feasible solutions.** Experiments show that EngiAgent greatly improves the consistency between numerical outputs and feasible solutions, ensuring that most generated results are valid. By contrast, existing methods often produce large gaps: for example, DS-Agent attains 62.26% numerical solutions on GPT-4o but only 5.66% are feasible; on DeepSeek-V3-671B it achieves 77.36% numerical solutions but just 28.30% feasible. MM-Agent shows smaller gaps but generates too few numerical results overall. EngiAgent (Coord.), however, maintains near consistency across all LLMs, with gaps as small as 1.89% on GPT-4o, 1.89% on Gemini-2.5 Flash, and 3.78% on DeepSeek-V3-671B, confirming that the vast majority of its numerical solutions satisfy real-world constraints.

**Contribution of the fully connected coordinator.** Ablation studies further confirm the importance of the fully connected coordinator. Replacing it with a fixed pipeline EngiAgent (Fixed) leads to consistent drops in both numerical solution rate and feasibility rate, with average declines of more than 10% across the three models. This demonstrates that flexible coordination is essential for robust feasibility in engineering tasks.

### 6.3 LIMITATIONS OF PURELY TEXT-BASED JUDGMENTS

In recent studies, text-based judgment has become a common approach to evaluating LLM performance on open-ended tasks. These methods, often powered by rubric-based scoring or LLM-as-a-judge frameworks, assign scores based on the linguistic quality, logical structure, and surface plausibility of model outputs. While convenient, such judgments are poorly aligned with the requirements of engineering problem solving, where feasibility is paramount.

As illustrated in Table 1 and Figure 4, text-based scores often appear inflated. Many solutions that are infeasible under engineering constraints nevertheless receive relatively high ratings because they are well-written or logically coherent at the textual level. This creates a misleading distribution in which feasible and infeasible solutions substantially overlap, obscuring the true capability gap. For example, solutions that violate energy conservation, exceed device capacity, or ignore operational safety limits may still obtain high textual scores, despite being unusable in practice.

This discrepancy reveals a limitation: purely text-based judgments reward expression rather than execution. They assess whether an answer looks convincing but cannot verify whether it actually satisfies numerical requirements, physical laws, or real-world operational constraints. Consequently, relying solely on such evaluations risks overestimating model performance in engineering domains. To address this issue, feasibility-based validation must be incorporated, ensuring that evaluation reflects not only linguistic plausibility but also practical implementability.

### 6.4 COST EFFICIENCY ANALYSIS

We further evaluate the runtime efficiency of different agents across three leading LLMs (Table 2), considering duration, token usage, and cost. Zero-shot prompting is the cheapest but has almost no feasible solutions, offering little practical value. Baseline agents such as DS-Agent and ResearchAgent incur relatively low costs but perform poorly in engineering feasibility. MM-Agent achieves stronger results in some dimensions but suffers from substantially higher inference overhead, with both runtime and token consumption at the highest level, resulting in poor overall cost-effectiveness.

In contrast, EngiAgent (Coord.) demonstrates consistently strong cost efficiency across all three LLMs. While its runtime and token usage are higher than those of some baseline agents, it achieves markedly higher feasibility rates while keeping inference costs within a reasonable range. The average

Table 2: Experimental results on runtime, token usage, and inference cost of different agents across three LLMs. Duration is measured in seconds, and tokens represent the total number of input and output tokens processed. Cost is estimated in USD based on the official pricing of each model: GPT-4o at $2.50 / 1M input tokens and $5.00 / 1M output tokens, Gemini-2.5 Flash at $0.30 / 1M input tokens and $2.50 / 1M output tokens, and DeepSeek-V3-671B at $0.27 / 1M input tokens and $1.10 / 1M output tokens.

| Methods | GPT-4o | | | Gemini-2.5 Flash | | | DeepSeek-V3-671B | | |
|---|---|---|---|---|---|---|---|---|---|
| | Duration (s) | Tokens | Cost ($) | Duration (s) | Tokens | Cost ($) | Duration (s) | Tokens | Cost ($) |
| Zero-shot | 112 | 6,986 | 0.02 | 46 | 20,767 | 0.04 | 110 | 7,893 | 0.01 |
| ResearchAgent | 306 | 181,324 | 0.63 | 85 | 37,134 | 0.04 | 218 | 14,943 | 0.01 |
| DS-Agent | 243 | 41,688 | 0.13 | 455 | 196,014 | 0.22 | 786 | 75,337 | 0.04 |
| MM-Agent | 1231 | 280,452 | 0.81 | 1714 | 648,042 | 0.44 | 1643 | 263,113 | 0.11 |
| EngiAgent (Fixed) | 629 | 154,352 | 0.47 | 1743 | 867,297 | 0.90 | 1650 | 198,657 | 0.09 |
| EngiAgent (Coord.) | 659 | 171,941 | 0.51 | 1119 | 813,157 | 0.67 | 1251 | 178,077 | 0.07 |

cost on GPT-4o is only $0.51, Gemini-2.5 Flash is $0.67, and on DeepSeek-V3-671B it is as low as $0.07, which is substantially lower than that of traditional multi-round frameworks. These results highlight that EngiAgent achieves a favorable balance between feasibility and efficiency.

A direct comparison with the baseline variant EngiAgent (Fixed) highlights the effectiveness of the Fully Connected Coordinator. While overall costs remain similar across all three LLMs, EngiAgent (Coord.) consistently achieves higher feasibility. On GPT-4o, feasibility increases by +16.98% at nearly the same cost ($0.51 vs. $0.47). On Gemini-2.5 Flash, it reduces both cost ($0.67 vs. $0.90) and tokens (813k vs. 867k) while improving feasibility by +11.32%. On DeepSeek-V3-671B, it further lowers cost ($0.07 vs. $0.09) and runtime (1251s vs. 1650s), with feasibility gains exceeding +7.55%. These results confirm that the fully connected coordinator substantially improves robustness and feasibility without introducing significant overhead.

## 6.5 ABLATION STUDY ON CORE COMPONENTS

To investigate the contribution of each module in EngiAgent, we conduct an ablation study on GPT-4o. Specifically, we analyze the effects of prompt engineering, the verifier, the fully connected coordinator, and the forced agent switching mechanism. As shown in Figure 5, we evaluate four degraded variants: (1) removing prompt engineering while keeping only minimal task instructions (w/o prompt eng.); (2) removing the verifier and relying solely on the remaining agents for modeling and solving (w/o verifier); (3) replacing the fully connected coordinator with a fixed sequential workflow (w/o fully connected coordinator, i.e., EngiAgent (Fixed)); and (4) disabling forced agent switching so that error correction relies only on the coordinator's internal reasoning (w/o forced switching).

Experimental results show that prompt engineering is important for accurate information extraction and modeling guidance, and its removal significantly reduces both the numerical-solution rate and feasibility. Removing the verifier leads to data inconsistencies and constraint omissions, causing a large decrease in feasibility and reproducing typical failure types discussed in Section 3. In addition, EngiAgent (Fixed) lacks the ability to dynamically route errors to appropriate agents, resulting in simultaneous degradation in both numerical outputs and feasibility, consistent with the observations reported in Section 6.2. Forced agent switching only serves as a last-resort mechanism triggered

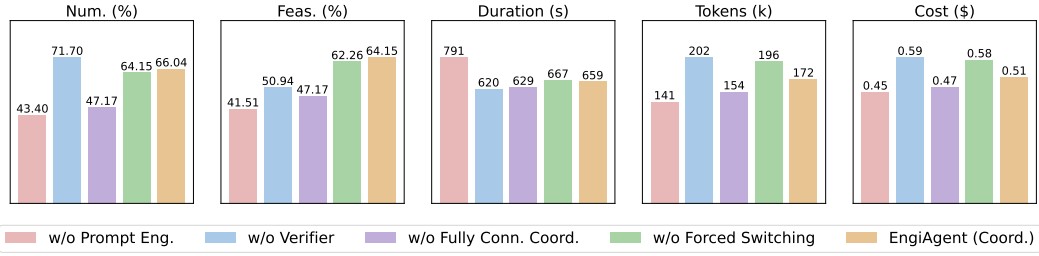

Figure 5: Ablation results over five metrics comparing degraded variants and the full EngiAgent.

Figure 6: An example output generated by EngiAgent. The problem statement is extracted from a power system market allocation task with multiple supply and demand units and transformed into a modeling input. The system automatically produces the corresponding Pyomo code snippet and solves it under given parameters to obtain the optimal results. The outputs report key power flow allocations and local generation across high and low scenarios, confirming feasibility and constraint satisfaction.

when repetitive debugging cycles emerge; disabling it does not noticeably affect solution quality, but slightly increases unnecessary retries and incurs additional computational overhead.

In summary, prompt engineering and the verifier provide essential modeling and feasibility checks, while the fully connected coordinator and forced switching improve adaptive routing and debugging efficiency, enabling EngiAgent to produce feasible solutions in open-ended engineering tasks.

### 6.6 EXAMPLE OUTPUT OF ENGIAGENT

To demonstrate the characteristics of EngiAgent's outputs, we present an example based on problem statement data (Figure 6). All results are strictly grounded in the numerical parameters provided in the problem. The system first transforms the natural language description into a standardized modeling format, then the Modeler automatically generates Pyomo code. Optimization is performed within this code to produce explicit numerical results. The final output not only provides concrete values for key variables but also confirms feasibility and constraint satisfaction.

## 7 CONCLUSION

This paper addresses the core challenge that LLMs often fail to guarantee feasibility in engineering problem solving. We propose EngiAgent, a multi-agent framework equipped with a fully connected coordinator that enables flexible feedback and targeted correction across problem analysis, modeling, verification, solving, and evaluation. This design substantially enhances robustness and ensures feasibility throughout the workflow. Experimental results demonstrate that EngiAgent significantly outperforms existing methods across multiple SOTA LLMs, achieving higher rates of feasible solutions and narrowing the gap between numerical outputs and practically valid results. Moreover, EngiAgent attains strong feasibility while maintaining reasonable efficiency and cost, highlighting its practicality and potential for broader application. Overall, this study establishes a new pathway toward feasibility-oriented intelligent systems for engineering applications.

## 8 ETHICS STATEMENT

This paper does not involve human subjects, personal data, or sensitive information. All datasets are sourced from publicly available engineering benchmarks and educational materials, and have been carefully reviewed to avoid privacy or security concerns. The work aims to improve the feasibility and reliability of LLM–based agents in engineering problem solving and does not produce harmful or unsafe applications. We acknowledge and adhere to the ICLR Code of Ethics, including principles of fairness, transparency, and research integrity.

## 9 REPRODUCIBILITY STATEMENT

We have made all resources necessary for reproducibility available. The complete implementation, including task pipelines, agent prompts, and evaluation scripts, is released in an anonymous repository: https://anonymous.4open.science/r/EngiAgent-1C8A/.

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

# Contents

## A  THE USE OF LARGE LANGUAGE MODELS

In this work, LLMs were used for grammar checking and language polishing during paper writing, and within the method they were invoked as intelligent agents in the EngiAgent framework to perform reasoning and modeling tasks.

## B  DATASET FOR ENGIAGENT EVALUATION

### B.1  CLASSIFICATION OF PROBLEM DOMAINS

The first category is *Market and Multi-Agent Decision-Making*. This category emphasizes the strategic interactions among multiple autonomous participants, with a particular focus on equilibrium modeling and competitive bidding. Typical tasks include electricity market bidding strategies, carbon market equilibrium analysis, and price forecasting under uncertainty.

The second category is *Scheduling and Resource Allocation*. This category centers on the optimization of resources, tasks, and time within a unified control framework. Representative examples involve production scheduling, task assignment, routing optimization, and unmanned vehicle mission allocation.

The third category is *Planning and Design*. This category addresses the optimization of layouts and facility locations for complex systems. Problems in this area include power grid expansion, infrastructure deployment, structural optimization, and urban pipeline planning.

The final category is *Control and Autonomous System Modeling*. This category highlights dynamic modeling and real-time feedback control of autonomous systems. Example tasks include UAV swarm

coordination, trajectory planning with obstacle avoidance, adaptive control under disturbances, and autonomous navigation.

## B.2 CATEGORY STATISTICS

To ensure comprehensive coverage of engineering challenges, the dataset is constructed with a balanced yet diverse distribution of problems. Table 3 summarizes the number of tasks in each category. In total, the dataset contains fifty-three tasks, distributed across four major categories.

Table 3: Statistics of problem categories in the dataset.

| Category | Number of Tasks |
|---|---|
| Market and Multi-Agent Decision-Making | 21 |
| Scheduling and Resource Allocation | 12 |
| Planning and Design | 10 |
| Control and Autonomous System Modeling | 10 |
| **Total** | 53 |

## B.3 DATASET FIELDS

Each problem entry is annotated with structured metadata to facilitate systematic evaluation. The fields and their descriptions are summarized in Table 4.

Table 4: Description of dataset fields.

| Field | Description |
|---|---|
| ID | Unique identifier for each problem. |
| Problem | Natural-language description of the problem. |
| Reference | Citation of the original paper or publication from which the problem is derived. |
| Field | Category label, chosen from the four domains in Section A.1. |
| Information Extraction (IE) | Ability to identify numerical values, constraints, and objectives. |
| Domain-Specific Reasoning (DR) | Ability to apply engineering knowledge and logical inference. |
| Multi-Objective Decision-Making (MO) | Ability to balance multiple objectives such as cost, efficiency, and reliability. |
| Uncertainty Handling (UH) | Robustness against perturbations, incomplete information, or stochastic variations. |
| Feasibility | Indicator of whether the solution not only satisfies all explicit constraints but also remains consistent with the numerical data and conditions stated in the problem description. |

## B.4 EVALUATION DIMENSIONS

The evaluation framework is designed to capture the multifaceted requirements of engineering problem solving. It consists of four primary dimensions and one final criterion:

- **Information Extraction (IE):** Measures whether the model can correctly identify essential parameters such as values, objectives, and constraints.

- **Domain-Specific Reasoning (DR):** Evaluates the ability to apply specialized engineering knowledge to derive intermediate models or formulations.

- **Multi-Objective Decision-Making (MO):** Assesses the capability of balancing multiple and potentially conflicting objectives.
- **Uncertainty Handling (UH):** Examines the robustness of solutions under incomplete or perturbed information.
- **Feasibility:** Serves as the ultimate criterion, determining whether solutions both satisfy all problem constraints and remain fully consistent with the given numerical data and conditions.

### B.5 FEASIBILITY EVALUATION PRINCIPLES

In engineering tasks, the core purpose of *Feasibility* is to determine whether a proposed solution violates essential physical, logical, or operational constraints such that the solution becomes impossible to implement in practice. Feasibility evaluation does not require exhaustive descriptive completeness; rather, it examines whether there exists any violation that fundamentally invalidates the solution. A solution is therefore considered feasible only if all necessary constraints are strictly satisfied. Conversely, any violation of these constraints immediately leads to infeasibility.

The feasibility evaluation principles can be summarized as follows:

- **Evaluation Objective:** Check whether the proposed solution violates key physical conservation laws, safety limits, logical consistency, or implementable operational requirements.
- **Definition of Violations:** Any condition that breaks energy conservation, capacity limits, safety constraints, logical consistency, or mathematical solvability is regarded as infeasible.
- **Not Part of Feasibility Assessment:** Whether certain internal quantities are explicitly modeled (e.g., whether an intermediate variable is recorded), whether full derivation steps are shown, or whether all implementation details are specified does not affect feasibility as long as all constraints remain satisfied.

Thus, the essence of feasibility evaluation is to determine whether any critical constraint is violated, rather than requiring the solution to exhaustively specify every modeling detail.

**Example.** Consider the case of photovoltaic (PV) curtailment. One can explicitly model the unused PV energy through a variable $cur_{i,t}$ that appears in the energy balance constraint, or implicitly represent curtailment by allowing $g_{i,t}^{\text{gen}}$ to operate below its maximum bound, i.e.,

$$0 \leq g_{i,t}^{\text{gen}} \leq \overline{g}_{i,t},$$

without outputting a specific curtailment value. Both approaches are physically valid because PV units can always reduce their power output.

However, the implications for feasibility verification differ:

- **Explicit modeling:** If $cur_{i,t}$ is explicitly introduced, it must participate in energy conservation verification together with generation, demand, charging/discharging, and trade flows. Any inconsistency between the recorded curtailment and the actual energy balance would constitute a constraint violation and must be classified as infeasible.
- **Implicit modeling:** If no explicit curtailment variable is used, no additional verification is required; feasibility remains guaranteed as long as the generation upper bound is satisfied and the energy balance accounts for actual generation.

Hence, feasibility hinges on strict satisfaction of constraints, while explicit or implicit representation of internal quantities reflects only differences in modeling granularity, not differences in feasibility conditions.

### B.6 ANNOTATION STANDARDS AND EXPERT DECISION PROTOCOL

To ensure the reliability, interpretability, and traceability of feasibility annotations in the EngiAgent dataset, we adopted a structured, fully human-driven labeling workflow involving more than 20 experienced domain experts. Each data sample was jointly evaluated by two annotators, who determined

feasibility and extracted the essential hard constraints strictly based on explicit problem statements and associated numerical information. When disagreement or uncertainty arose, a third senior annotator with deeper domain experience (all holding a PhD degree) adjudicated the final conclusion to ensure consistency and reduce subjective interpretation bias. When necessary, annotators were allowed to reference fundamental physical laws, established engineering practices, and standard operational rules; however, personal modeling preferences or additional subjective assumptions were strictly prohibited to maintain objectivity and real-world applicability.

Importantly, feasibility labels were **not** generated automatically by LLMs. No LLM-based automated labeling mechanism was used. LLM tools were permitted only for auxiliary clarification tasks, such as terminology interpretation, and their outputs were **never** used as direct feasibility decisions or replacements for human judgment. All feasibility assessments were made solely by human experts based on the task description and explicit engineering constraints, independent of any model-generated solutions, ensuring a fully human-controlled and system-agnostic evaluation process.

To promote consistent reasoning and avoid divergence in interpretation, annotators were provided with a reference prompt serving solely as a cognitive guide rather than a decision rule or automated inference template. The reference prompt is as follows:

---

**Reference Prompt for Feasibility Assessment**

Given the modeling problem, identify the essential constraints that must be satisfied to ensure feasibility. Focus only on the most critical mandatory conditions implied by the problem statement, taking into account both explicit limitations and general engineering knowledge or operational rules that must be respected. Present the constraints in a numbered list (1., 2., 3., . . . ), using clear and concise reasoning.

---

This protocol ensures that feasibility annotations are grounded in rigorous engineering reasoning and maintain transparency, consistency, and reproducibility for future research and dataset reuse.

### B.7 EXAMPLE ADJUDICATION AND ERROR ANALYSIS

**Problem Description.** In a simplified electricity market, two conventional generators, one wind farm, and one storage unit jointly supply demand over eight hourly periods. The conventional generators have block-based generation costs and ramping limits, while the wind farm provides zero-cost and time-varying output. The storage unit can charge and discharge subject to efficiency losses and limited energy capacity. Market clearing matches supply and demand, with prices determined by the marginal offer. The objective is to formulate a mathematical optimization model that determines hourly dispatch, storage scheduling, and resulting prices, and then analyze the effects of storage on (i) social welfare, (ii) wind curtailment, and (iii) firm profits.

| Unit | Block Cap. (MW) | Ramp Limit (MW/h) | Cost ($/MWh) | Notes |
|---|---|---|---|---|
| Conventional Gen 1 | 120, 120 | +200 / −180 | 30, 40 | Inc. marginal cost |
| Conventional Gen 2 | 80, 80 | +150 / −130 | 50, 60 | Inc. marginal cost |
| Wind Farm | ≤250 (vary by hour) | N/A | 0 | Hourly avail. provided |
| Storage Unit | 200 MW (charge/discharge) | – | 0 | $SOC_0 = 100$, cap=200 MWh, $\eta = 0.85$ |

**System Data.** Hourly demand $D_t$ and wind availability $\overline{W}_t$ are explicitly provided for $t = 1, \ldots, 8$.

**Feasibility Constraint Set.** The following six constraints are annotated as **required hard feasibility conditions**:

1. Hourly power balance (for $t = 1, \ldots, 8$):
$$W_t + G_t^{(1)} + G_t^{(2)} + H_t = D_t + C_t.$$

2. Conventional generator block limits:
$$G_t^{(1)} = G_{t,1}^{(1)} + G_{t,2}^{(1)}, \quad 0 \le G_{t,1}^{(1)} \le 120, \ 0 \le G_{t,2}^{(1)} \le 120,$$
$$G_t^{(2)} = G_{t,1}^{(2)} + G_{t,2}^{(2)}, \quad 0 \le G_{t,1}^{(2)} \le 80, \ 0 \le G_{t,2}^{(2)} \le 80.$$

3.  Conventional generator ramping limits (for $t = 2, \ldots, 8$):

$$G_t^{(1)} - G_{t-1}^{(1)} \leq 200, \qquad G_{t-1}^{(1)} - G_t^{(1)} \leq 180,$$
$$G_t^{(2)} - G_{t-1}^{(2)} \leq 150, \qquad G_{t-1}^{(2)} - G_t^{(2)} \leq 130.$$

4.  Wind availability bound:

$$0 \leq W_t \leq \overline{W}_t.$$

5.  Storage SOC dynamics, bounds, and initial condition:

$$SOC_t = SOC_{t-1} + \eta_c C_t - \frac{1}{\eta_d} H_t, \qquad 0 \leq SOC_t \leq 200,$$
$$SOC_0 = 100.$$

6.  Storage power limits and charge/discharge exclusivity:

$$0 \leq C_t \leq 200 u_t, \quad 0 \leq H_t \leq 200 v_t, \quad u_t + v_t \leq 1, \quad u_t, v_t \in \{0, 1\}.$$

**Adjudication and Error Analysis.** All three annotators agreed that the above six constraints are mandatory feasibility conditions because they encode (i) energy conservation, (ii) device capability limits, and (iii) physically valid storage dynamics. During the annotation process, a disagreement occurred on whether a *terminal SOC condition* (e.g., $SOC_8 = SOC_0$ or $SOC_8 \geq SOC_0$) must also be included as a hard requirement. One annotator initially rated the problem as *infeasible* without such a terminal constraint, arguing that discharging to exhaustion may be undesirable in long-term planning. This created a potential **false-infeasible** labeling case.

After expert adjudication, it was clarified that terminal SOC requirements reflect *modeling policy or preference* rather than *physical feasibility necessity*, because the problem statement does not specify an operational-horizon sustainability requirement. Therefore, as long as Constraints (1)–(6) are satisfied, the problem admits a valid feasible solution space. The final feasibility label for P1 is recorded as **Feasible**. This example is documented as a representative case where over-restrictive assumptions could incorrectly induce a false-infeasible judgment.

### B.8  COMPLEXITY DISTRIBUTION OF ENGINEERING TASKS

To further clarify the complexity of the engineering tasks used in EngiAgent, we report the empirical distribution of constraint and parameter scales over all tasks. As discussed in the main text, these tasks are open-ended: the benchmark specifies engineering objectives and physical or operational constraints, while the exact mathematical formulation (including the number of variables, the number of constraints, and solver class) is determined by how the LLM chooses to model each problem. Thus, complexity is not a predefined attribute of the task description, but an emergent property of the generated models.

Figure 7 shows the distributions of the number of constraints and the number of parameters across all generated solutions. Both metrics exhibit a clear long-tailed pattern: most tasks fall into a medium scale range (approximately 5–20 constraints), while a non-negligible subset reaches substantially higher complexity (e.g., more than 50 constraints or more than 100 parameters). This behavior is consistent with real-world engineering practice, where routine operational problems coexist with a smaller number of highly constrained, large-scale planning or coordination tasks. These results support our claim that the benchmark covers a broad spectrum of engineering difficulty, rather than a narrow set of simplified toy problems.

## C  CORE PROMPT DESIGN OF ENGIAGENT

The effectiveness of EngiLLM relies heavily on carefully crafted prompts instructing each specialized agent to perform domain-specific tasks with precision and consistency. These prompts incorporate engineering domain knowledge, structured output formats, and error handling mechanisms to ensure reliable automated modeling and solving workflows. Each agent's prompt is designed to maintain coherence with the overall system architecture while addressing the unique requirements of their respective roles in the engineering problem-solving pipeline.

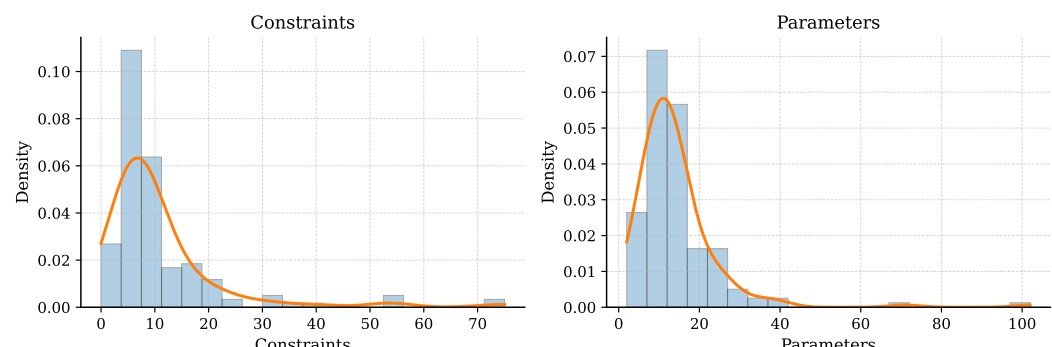

Figure 7: Distribution of task complexity in the EngiAgent benchmark. Left: density distribution of the number of constraints across all 53 tasks. Right: density distribution of the number of parameters. Both exhibit long-tailed behavior, indicating coverage from medium-scale to highly constrained engineering problems.

## C.1 ANALYZER

The Analyzer prompt is designed to systematically extract and structure engineering problem information from natural language descriptions. It guides the LLM to identify decision variables, constraints, objectives, and implicit engineering rules while maintaining consistency with downstream modeling requirements. The prompt emphasizes comprehensive problem understanding and standardized JSON output generation to ensure seamless integration with the Modeler.

```
1 initial_modeling_analysis_prompt = """You are a senior engineering
      problem modeling expert. Please perform a **high-quality systematic
      analysis** of the following engineering problem, referring to the
      retrieved modeling knowledge to extract complete modeling elements.
2
3 **Problem Description:**
4 {problem_text}
5
6 **Retrieved Relevant Modeling Methods:**
7 - **Recommended Domain:** {hmml_analysis.get('domain', 'General
      Engineering')}
8 - **Subdomain:** {hmml_analysis.get('subdomain', 'Optimization')}
9 - **Recommended Method:** {hmml_analysis.get('method', 'Mathematical
      Programming')}
10 - **Confidence:** {hmml_analysis.get('confidence', 0.5):.2f}
11
12 **Modeling Guidance:**
13 - **Modeling Approach:** {modeling_guidance.get('approach', 'Mathematical
       optimization-based modeling')}
14 - **Core Concept:** {modeling_guidance.get('methodology', 'Construct
      objective function and constraints')}
15 - **Mathematical Framework:** {modeling_guidance.get('framework', '
      Optimization framework')}
16 - **Solution Strategy:** {modeling_guidance.get('solution_strategy', '
      Numerical optimization methods')}
17
18 **Application Context:**
19 - **Typical Applications:** {hmml_result.get('application_context', {}).
      get('applications', 'Engineering optimization problems')}
20 - **Method Advantages:** {hmml_result.get('application_context', {}).get
      ('advantages', 'Mature theory')}
21
22 ### Role and Mission ###
23 You are a top-tier "system architect" in charge of designing a complex,
      open-ended engineering problem into a **highly structured, traceable,
       and hierarchical JSON modeling blueprint**. This blueprint is the
```

```
     sole basis for downstream code generation agents to build accurate,
     feasible solving models.

### Core Thinking Framework: Four Modeling Dimensions ###
When analyzing a problem, you must consider the following four dimensions
     as implicit indicators of your analysis. Your final output structure
     must reflect the results of this deep thinking process, but **do not
     ** create top-level blocks in the JSON for these four dimensions.
1.  **Information Extraction (Information Extraction)**: Your goal is to
    identify all "atomic" information needed for modeling - entities,
    numerical values, relationships, and goals. This is the foundation of
     modeling.
2.  **Domain-specific Reasoning (Domain-specific Reasoning)**: You need
    to "dress" these "atomic" pieces of information with "domain clothing
    ". Using professional engineering knowledge, transform raw data into
    meaningful parameters, convert relationships into physical or logical
     constraints, and select the most appropriate modeling paradigm.
3.  **Multi-objective Decision-making (Multi-objective Decision-making)
    **: You need to prioritize all goals. A **core optimization goal**
    must be clearly defined, while other goals are recognized as
    secondary or trade-off items. This is the "compass" guiding your
    decisions.
4.  **Uncertainty Handling (Uncertainty Handling)**: You need to identify
     all "cracks" in the information - missing, fuzzy, or variable parts.
     Your task is to "fill" these cracks through reasonable assumptions,
    ensuring the integrity and certainty of the core model.

### Strict Output Format: Hierarchical JSON Modeling Blueprint ###
You must, and can only output a JSON object wrapped in ```json...```.
     This structure has been carefully designed to ensure the completeness
     , hierarchy, and executability of downstream agents.
```json...
{
  "modeling_context": {
    "problem_essence": "Summarize the core engineering optimization
    problem in one sentence, formatted as 'Under [Key Constraints],
    optimize [Decision Variables] to achieve [Core Goals].".
    "engineering_domain": "Specific engineering field (e.g., 'Power
    System Dispatch', 'Supply Chain Network Design', 'Aerospace Planning
    ')",
    "modeling_paradigm": "Recommended core modeling methods based on
    problem characteristics and industry conventions (e.g., 'Mixed-
    Integer Linear Programming', 'Quadratic Constraint Programming', '
    Stochastic Optimization')",
    "solution_scope": "Clearly define the core scope and boundary
    conditions of this modeling exercise."
  },
  "core_model_elements": {
    "description": "Defines the minimum viable model of the problem (
    Minimum Viable Model), including all necessary elements for
    constructing a basic solution method. This part must be self-
    consistent and complete, directly usable for code implementation.",
    "decision_variables": [
      {
        "name": "Standard Mathematical Symbol Variable Name (e.g., P_g_t)
    ",
        "description": "Exact engineering meaning of the variable, index
    definition (e.g., 'g' represents generating units, 't' represents
    time periods) and unit",
        "type": "continuous/integer/binary",
        "domain": "Mathematical definition domain (e.g., '>= 0', '[0,
    1]')",
        "shape": "Dimension or size of the variable (e.g., '[G, T]' G is
    the number of units, T is the number of time periods)"
      }
```

```
52        ],
53        "parameters": [
54          {
55            "param_id": "PARAM_01",
56            "name": "Mathematical Symbol of Parameter (e.g., C_g_max)",
57            "value": "Specific numerical values or complete arrays/lists from
          the original problem description. No omissions or descriptive
          substitutions are allowed.",
58            "unit": "Physical unit (e.g., 'MW', 'USD/MWh', 'kg')",
59            "description": "Exact engineering meaning and index explanation
          of the parameter.",
60            "source_reference": "Indicates the source or basis of this data
          in the original problem description (e.g., 'from Table 1, Column 3',
          'Section 5 of the problem description explicitly states')"
61          }
62        ],
63        "objective_function": {
64          "name": "Name of the core objective function (e.g.,
          TotalOperatingCost)",
65          "type": "minimize/maximize",
66          "expression": "Complete mathematical expression using the defined
          variable and parameter symbols.",
67          "components": [
68            {
69              "component_expr": "One part of the expression (e.g., sum(c_g *
          P_g_t for g, t))",
70              "description": "Engineering meaning of this component (e.g., '
          Total Fuel Cost')"
71            }
72          ]
73        },
74        "constraints": [
75          {
76            "constraint_id": "CONST_01",
77            "name": "Unique Identifier of Constraint (e.g., 'PowerBalance')",
78            "expression": "Complete mathematical (equation/inequality)
          expression using the defined variable and parameter symbols.",
79            "category": "Category of the constraint ('Physical Laws', '
          Resource Capacity', 'Supply-Demand Balance', 'Operational Logic', '
          Strategic Requirements')",
80            "description": "Engineering significance and importance of the
          constraint."
81          }
82        ]
83      },
84      "extended_analysis_and_robustness": {
85        "description": "Includes supplementary, expanded, and considerations
        under uncertainty for the core model, which is key to advanced
        analysis and ensuring robustness of solutions.",
86        "key_assumptions": [
87          {
88            "assumption_id": "ASSUM_01",
89            "content": "Explicit assumptions made to handle missing
          information or simplify the model.",
90            "justification": "Engineering principles, industry conventions,
          or logical basis for this assumption.",
91            "impact_on_model": "Explicitly states how this assumption
          specifically affects a specific element in `core_model_elements` (e.g
          ., 'Setting the value of PARAM_05 `line_efficiency` to 0.98', '
          Simplifying the calculation formula of CONST_03 `PowerFlow`')"
92          }
93        ],
94        "uncertainty_sources": [
95          {
96            "source_id": "UNCERT_01",
```

```
          "description": "Key sources of uncertainty identified (e.g., '
      Future Market Electricity Price', 'Equipment Failure Rate')",
          "affected_elements": ["IDs of parameters or variables affected by
       this uncertainty (e.g., 'PARAM_10')"],
          "handling_strategy": "Recommended strategies for handling (e.g.,
      'Using Expected Values for Deterministic Modeling', 'Conducting
      Sensitivity Analysis', 'Using Scenario Analysis Approach')
        }
      ],
      "trade_off_analysis": {
        "secondary_objectives": [
          {
            "name": "Secondary or potential optimization goals (e.g., '
      MinimizeCarbonEmissions')",
            "expression": "Mathematical expression.",
            "conflict_with": "Which core or secondary goals are in tension
      with each other (e.g., 'TotalOperatingCost')"
          }
        ],
        "soft_constraints": [
          {
            "name": "Soft constraints or preferences (e.g., '
      PreferredMaintenanceWindow')",
            "description": "Conditions hoped for but not necessarily
      required, often modeled by adding penalty terms to the objective
      function."
          }
        ]
      },
      "sensitivity_factors": [
        {
          "param_id": "ID of key parameters for conducting sensitivity
      analysis (from `parameters` section)",
          "justification": "Why this parameter might have a significant
      impact on the model results."
        }
      ]
  }
}
```

### Golden Command and Final Verification ###
1.  **Absolute Data Integrity**: Every valid numerical value in the
    original problem **must** be present in the `parameters` list with a
    unique entry, and its origin must be traceable through the `
    source_reference` field.
2.  **Mandatory Internal References**: Cross-references within JSON are
    mandatory. For example, `sensitivity_factors` must reference `
    param_id` of `parameters`; `key_assumptions`'s `impact_on_model` must
     explicitly point to a specific element in `core_model_elements`.
    This ensures traceability and consistency of the model.
3.  **Core Model Priority**: `core_model_elements` is the cornerstone. It
     must be a model that can be independently solved and clearly defined
    . All supplementary, trade-off, and uncertainty analyses are based on
     it in `extended_analysis_and_robustness`.
4.  **Clear Roles and Responsibilities**: Strictly differentiate between
    `core_model_elements` (what they do) and `
    extended_analysis_and_robustness` (how to do it better/more robustly)
    . Do not mix in assumptions or uncertain elements in the core model.
5.  **Designed for Code Generation**: All mathematical expressions (`
    expression`) must use standard, unambiguous mathematical notation so
    that downstream agents can easily parse and convert them into code.
    """

## C.2 MODELER

The Modeler prompt focuses on translating structured problem formulations into executable Pyomo optimization code. It employs domain-specific modeling patterns, syntax validation guidelines, and common pitfall avoidance strategies tailored for engineering optimization open questions. The prompt ensures mathematical rigor while maintaining code clarity and computational efficiency for various problem types including linear, mixed-integer, and nonlinear programming.

```
basic_model_code_generation_prompt = f"""### Role & Mission ###
You are a top-tier code generation engine specialized in translating
    structured engineering modeling blueprints (JSON) into executable,
    industrial-grade Python code.

### Core Instructions ###
Your **ONLY** task is to generate complete, accurate, directly executable
     Python solving code based on the `MODELING_BLUEPRINT` JSON provided
    below. Every key-value pair in the JSON is a mandatory instruction
    that must be strictly followed.

### Input: Modeling Blueprint (MODELING_BLUEPRINT) ###
```json
{analysis_result_str}
```

### CRITICAL Data Structure & Dimension Rules ###
**NEVER use placeholder data like [...] or ellipsis**
**Always provide complete, specific data arrays**
- If JSON contains "[...]" or "...", replace with realistic example data
- Match array dimensions exactly to variable dimensions
- For time series: use 1D arrays like [0.1, 0.15, 0.2, ...]
- For scenarios: create 2D arrays like [[value1, value2], [value3, value4
    ], ...]

**NEVER mismatch array dimensions with variable indices**
**Ensure parameter arrays match variable indexing**
```python
# Wrong: Using 1D array with 3D variables
Electricity_price_t = [0.1, 0.15, ...]  # 1D
model.P_ev_t_s[e, t, s]  # 3D - MISMATCH!

# Correct: Match dimensions or simplify variables
# Option 1: Simplify to 2D variables
model.P_ev_t[e, t]  # 2D
Electricity_price_t = [0.1, 0.15, ...]  # 1D - MATCH!

# Option 2: Create appropriate multi-dimensional data
Electricity_price_t_s = [[0.1, 0.12], [0.15, 0.17], ...]  # 2D
model.P_ev_t_s[e, t, s]  # 3D with proper indexing
```

### CRITICAL Pyomo Summation Rules ###
**NEVER use summation() from pyomo.environ - it's error-prone**
**Always use Python's built-in sum() with generator expressions**
```python
# Wrong: Using summation() function
from pyomo.environ import summation
electricity_cost = summation(Electricity_price_t, model.P_ev_t)  # ERROR!

# Correct: Using sum() with proper indexing
electricity_cost = sum(Electricity_price_t[t-1] * model.P_ev_t[e, t]
                    for e in E for t in T)
```

**NEVER pass arrays directly to sum() - always use explicit indexing**
**Always iterate over indices and access array elements**
```

```python
# Wrong: Direct array operations in sum
sum(price_array * model.var for ...)  # ERROR!

# Correct: Index-based access
sum(price_array[i] * model.var[i] for i in range(len(price_array)))
```

### CRITICAL Pyomo Syntax Rules ###
**NEVER use Python built-ins in constraints**: max(), min(), abs()
**NEVER use if statements with Pyomo variables**: if model.x[i] >= 0: ...
**NEVER compare Pyomo variables in boolean context**: if P_ev_t[ev, t] >= 0
**NEVER use ternary operators with Pyomo variables**: value = model.x[i] if model.x[i] >= 0 else 0
**NEVER mix Pyomo expressions with conditional logic**: model.x[i] * (1 if condition else 0)
**For conditional logic**: Use separate variables or Pyomo Piecewise functions
**For max constraints**: Use auxiliary binary variables with Big-M method
**For absolute value**: Use two inequality constraints: x >= y and x >= - y

### MOST COMMON ERRORS TO AVOID ###
**"Cannot convert non-constant Pyomo expression to bool" – CAUSED BY:**
```python
# WRONG - These will cause fatal errors:
if model.P_s_t[t] >= 0:
    return model.P_s_t[t] /
value = model.x[i] if model.x[i] > 0 else model.x[i] * 0.9
model.constraint = Constraint(expr=model.var[i] >= (5 if condition else 3))
```

**CORRECT alternatives:**
```python
# Method 1: Separate variables
model.P_charge = Var(T, domain=NonNegativeReals)
model.P_discharge = Var(T, domain=NonNegativeReals)

# Method 2: Binary variables with constraints
model.mode = Var(T, domain=Binary)
model.cons1 = Constraint(T, rule=lambda m,t: m.P[t] <= M * m.mode[t])
model.cons2 = Constraint(T, rule=lambda m,t: m.P[t] >= -M * (1 - m.mode[t]))
```

### CRITICAL: NEVER GENERATE INCOMPLETE CODE ###
**ABSOLUTELY FORBIDDEN**: Any code containing only "results = []" or similar empty placeholders
**ABSOLUTELY FORBIDDEN**: Comments like "# The rest of the code remains unchanged"
**ABSOLUTELY FORBIDDEN**: Any response that does not contain a COMPLETE Pyomo model

**MANDATORY REQUIREMENT**: Every response MUST contain a FULLY FUNCTIONAL Pyomo optimization model that can be executed immediately.

**Your code MUST include ALL of these components (NO EXCEPTIONS):**
1. Complete imports: `from pyomo.environ import *`
2. All parameter data with realistic values (no [...] placeholders)
3. Complete ConcreteModel() definition with ALL variables from JSON
4. Complete Objective function (maximize/minimize something meaningful)
5. ALL constraint definitions from the JSON properly implemented
6. Complete solver setup and solve() call

```
106  7. Result extraction and printing
107
108  **VERIFICATION CHECKLIST - Your code must pass ALL these checks:**
109  Does it import pyomo.environ?
110  Does it create a ConcreteModel()?
111  Does it define ALL variables mentioned in the JSON?
112  Does it implement ALL constraints from the JSON?
113  Does it define a meaningful objective function?
114  Does it call a solver and get results?
115  Is it longer than 100 lines of actual code?
116
117  ### Output Requirements ###
118  - Format: Output only one complete Python code block wrapped in ```python
         ...```
119  - Content: Code must include all necessary library imports, parameter
         definitions, model construction, solver calls, and clear result
         output
120  - Quality: Code must be robust with basic solver status checks
121  - No extra explanation: Except for the code itself and necessary comments
         , do not add any preamble or summary
122  - Solver setup: Include intelligent solver selection (glpk, cbc, ipopt)
         with timeouts and error handling
123  - Data integrity: All parameter arrays must be complete with realistic
         values, no placeholders
124  """
```

## C.3 VERIFIER

The Verifier prompt establishes comprehensive quality assurance protocols for semantic verification between problem descriptions and generated models. It guides the LLM to perform multi-dimensional consistency checks, identify logical inconsistencies, and provide detailed feedback for model refinement. The prompt emphasizes distinguishing between critical errors and acceptable simplifications while maintaining alignment with original engineering problem requirements. Notably, the prompt integrates a dynamic tolerance adjustment mechanism that automatically relaxes verification standards when consecutive verification failures are detected, promoting system convergence while preventing inefficient excessive retry loops.

```
1  # Dynamically adjust Verification tolerance
2              tolerance_adjustment = ""
3              if self.consecutive_verification_failures > 8:
4                  tolerance_adjustment = """
5  ** Special Notice: Multiple verification loops detected, please adopt
         more lenient verification standards**
6  - For complex game theory/bilevel optimization problems, allow reasonable
         modeling simplification
7  - For MPEC/KKT conditions, accept pragmatic-oriented approximate
         implementations
8  - Focus on verifying core logic correctness, be tolerant of technical
         details
9  - Only judge as mismatch when fundamental errors exist"""
10              elif self.consecutive_verification_failures > 5:
11                  tolerance_adjustment = """
12  **Notice: Verification difficulties detected, please appropriately relax
         verification standards**
13  - Maintain understanding and tolerance for modeling strategy differences
14  - Focus on substantial issues, ignore technical implementation details"""
15
16              prompt = f"""### Role and Mission ###
17  You are a top-tier, **extremely pragmatic** engineering system
         verification expert. Your core mission is to serve as the system's "
         gatekeeper". Your task is not to conduct line-by-line code auditing,
         but to judge from a **system modeling perspective** whether the
         provided Python code faithfully solves the original engineering
```

```
       problem in terms of **core logic and key objectives**. You must
       distinguish between **modeling strategy differences** and **
       substantial logical errors**.
18 {tolerance_adjustment}
19
20 Your primary principle is **"Promote Convergence"**:
21 - **Default Trust**: Unless the code contains **catastrophic, directional
       ** errors, it should default to `PASS`.
22 - **Understand Compromise**: Must recognize that the current code may be
       a **necessary compromise or simplification** made to solve **previous
        failures (such as overly strict constraints, solver infeasibility,
       etc.)**.
23 - **Minimal Intervention**: Your role is to confirm the overall direction
        is correct and the core has not deviated, not to micromanage
       technical details.
24
25 ### Input ###
26 1.  **[Problem Specification] Original Engineering Problem (The Problem
       Specification)**:
27      ```
28      {safe_original_problem}
29      ```
30 2.  **[Solution Implementation] Engineering Model Code (The Implemented
       Solution)**:
31      ```python
32      {safe_pyomo_code}
33      ```
34
35 ### Core Verification Instructions: Judgment Hierarchy Based on
       Historical Context ###
36 Please strictly follow this judgment logic to ensure your verification is
        context-aware.
37
38 **Step 1: Check for "Catastrophic" Errors (Deal-Breaker Check)**
39 This is the only reason you can judge as `FAIL`. Check if any of the
       following situations exist:
40 1.  **Objective Direction Reversed**: Minimization problems implemented
       as maximization, or vice versa.
41 2.  **Core Physical/Economic Law Violations**: Complete omission of
       constraints that define the problem foundation, such as "supply-
       demand balance", "energy conservation", etc.
42 3.  **Key Decision-Making Entities Missing**: In an obvious multi-agent
       game problem, the code completely fails to reflect interactions or
       decisions between different entities (even in simplified form).
43
44 *   If any of the above is found, immediately judge as `FAIL` and stop
       subsequent checks.
45 *   If none are found, **proceed directly to Step 2 and finally judge as
       `PASS`**.
46
47 **Step 2: Identify and Acknowledge "Pragmatic Deviations" (Pragmatic
       Deviation Recognition)**
48 When code passes Step 1 checks, it will be judged as `PASS`. Your task is
        to identify differences between the current code and the original
       problem specification, and judge whether this is a reasonable
       response to errors in the `[Historical Context]`.
49
50 Common **reasonable deviations that should be accepted as `PASS`**
       include:
51 -   **Relaxing/removing constraints to solve infeasible problems**:
       Should consider that previous reasonable modeling might have been "
       infeasible", and the current code relaxing or removing certain strict
        constraints is a **positive, should-be-encouraged** moderate
       compromise.
```

```
52  -    **Simplifying models to address LLM capability limitations or solving
         complexity**: Simplifying complex bilevel optimization, MPEC or
       equilibrium models to appropriate single-level optimization, as long
       as key economic incentives or trade-off relationships are embodied
       through parameters or proxy constraints, should be accepted.
53  -    **Technical adjustments**: Adjusting variable boundaries, parameter
       units, Big-M coefficients, solver options, etc., to improve numerical
         stability or find feasible solutions.
54  -    **Minor discrepancies in data or indices**: Should be ignored as long
       as they don't affect the core model logic and magnitude.
55
56  ### Output Format ###
57  Please strictly return your judgment in the following JSON format,
       wrapped with ```json...```. **When the model adopts "acceptable
       approximation", please provide minimal incremental correction
       suggestions in `suggestion` rather than outright rejection.**
58
59  ```json
60  {{
61    "mismatch_detected": true/false,
62    "mismatch_reason": "Specific mismatch reason",
63    "suggestion": "Fix or optimization suggestions",
64    "confidence": 0.0-1.0
65  }}
66  ```
67  """
```

# D  CASE STUDY

---

### Example: Rooftop PV Layout Optimization (P35)

PROBLEM DESCRIPTION

We consider a rooftop photovoltaic (PV) layout optimization problem. The objective is to minimize the total capital expenditure (Capex) while ensuring that the annual electricity demand $D_{\text{tar}} = 50{,}000$ kWh/yr is satisfied. The system consists of a single flat roof with usable area $A_{\text{use}} = f_{\text{use}} \cdot A_{\text{roof}} = 420$ m$^2$. All panels must share one common tilt–azimuth configuration, chosen from a small set of candidates.

DATA

The parameters of the rooftop system, available PV panel types, and candidate configurations are summarized in Tables 5 and 6.

Table 5: PV panel specifications.

| Panel Type | $P_i$ (kWp) | $a_i$ (m$^2$) | $c_i$ (€/panel) |
|---|---|---|---|
| Canadian 275 W | 0.275 | 1.64 | 130 |
| Jinko 330 W | 0.330 | 1.94 | 155 |
| Canadian 330 W | 0.330 | 1.94 | 175 |

Table 6: Candidate tilt–azimuth configurations and yields.

| Config $k$ | Tilt (°) | Azimuth (°) | $y_k$ (kWh/kWp·yr) |
|---|---|---|---|
| 1 | 20 | 160 | 1350 |
| 2 | 30 | 180 | 1450 |
| 3 | 35 | 135 | 1400 |

MODEL FORMULATION

**Decision Variables.**

- $n_i \in \mathbb{Z}_{\geq 0}$: number of panels of type $i$,
- $z_k \in \{0, 1\}$: binary variable for configuration choice,
- $K \geq 0$: installed capacity (kWp),
- $E_k \geq 0$: annual energy (kWh) for configuration $k$.

**Objective.**

$$\min \quad \text{Capex} = \sum_{i=1}^{3} c_i n_i + \varepsilon \sum_{i=1}^{3} n_i,$$

where $\varepsilon \ll \min_i c_i$ is a tie-breaker favoring fewer panels.

**Constraints.**

$$\sum_{k=1}^{3} z_k = 1, \qquad z_k \in \{0, 1\} \quad \text{(Single configuration)} \tag{1}$$

$$K = \sum_{i=1}^{3} P_i n_i, \qquad n_i \in \mathbb{Z}_{\geq 0} \quad \text{(Installed capacity)} \tag{2}$$

$$E_k \geq D_{\text{tar}} \cdot z_k, \qquad \forall k \quad \text{(Demand satisfaction)} \tag{3}$$

$$\sum_{i=1}^{3} a_i n_i \leq A_{\text{use}}, \qquad \text{(Usable roof area)} \tag{4}$$

$$E_k \leq y_k K + M(1 - z_k), \qquad \forall k \quad \text{(Linearized yield coupling)} \tag{5}$$

Optional constraints (not used in this case) include:

$$\sum_{i=1}^{3} a_i n_i \geq \alpha_{\min} A_{\text{use}}, \quad \alpha_{\min} \in [0, 1] \quad \text{(Coverage lower bound)} \tag{6}$$

$$n_i \leq \overline{n}_i, \quad i = 1, 2, 3 \quad \text{(Panel availability limit)} \tag{7}$$

OPTIMAL SOLUTION

The MILP was solved using GLPK. The optimal feasible solution is:

- Selected configuration: $k = 2$ (Tilt $30°$, Azimuth $180°$),
- Panel counts: $n_1 = 3$, $n_2 = 102$, $n_3 = 0$,
- Installed capacity: $K = 34.49$ kWp,
- Annual energy yield: $E = 50{,}000$ kWh,
- Capital expenditure: €16,200.

PERFORMANCE METRICS

$$S = p_{\text{el}} \cdot E = 9{,}000/\text{yr}, \qquad \text{Payback} = \frac{\text{Capex}}{S} \approx 1.8 \text{ years},$$

$$\text{CO}_{2,\text{avoided}} = \gamma_{\text{CO2}} \cdot E = 12{,}500 \text{ kg/yr}, \qquad \text{Roof coverage} = \frac{\sum_i a_i n_i}{A_{\text{use}}} \approx 49.6\%.$$

PYOMO IMPLEMENTATION

```python
from pyomo.environ import *

# Initialize the Pyomo model
model = ConcreteModel()

```

```python
6  # Define sets
7  model.I = RangeSet(3)   # panel types
8  model.K = RangeSet(3)   # configurations
9
10 # Parameters
11 params = {
12     'A_roof': 600,
13     'f_use': 0.7,
14     'A_use': 420,
15     'p_el': 0.18,
16     ' _CO2 ': 0.25,
17     'D_tar': 50000,
18     'P_i': [0.275, 0.33, 0.33],
19     'a_i': [1.64, 1.94, 1.94],
20     'c_i': [130, 155, 175],
21     'y_k': [1350, 1450, 1400],
22     '  ': 1e-05,
23     'M': 100000
24 }
25
26 # Variables
27 model.n_i = Var(model.I, domain=NonNegativeIntegers)
28 model.z_k = Var(model.K, domain=Binary)
29 model.K_installed = Var(domain=NonNegativeReals)
30 model.E_k = Var(model.K, domain=NonNegativeReals)
31
32 # Objective
33 model.MinimizeCapex = Objective(
34     expr=sum(params['c_i'][i-1]*model.n_i[i] for i in model.I) +
35           params['  ']*sum(model.n_i[i] for i in model.I),
36     sense=minimize
37 )
38
39 # Constraints
40 def energy_demand_rule(m, k):
41     return m.E_k[k] >= params['D_tar'] * m.z_k[k]
42 model.EnergyDemand = Constraint(model.K, rule=energy_demand_rule)
43
44 model.RoofArea = Constraint(
45     expr=sum(params['a_i'][i-1]*model.n_i[i] for i in model.I) <=
46     params['A_use']
47 )
48 model.ConfigurationChoice = Constraint(
49     expr=sum(model.z_k[k] for k in model.K) == 1
50 )
51
52 model.InstalledCapacity = Constraint(
53     expr=model.K_installed == sum(params['P_i'][i-1]*model.n_i[i]
54     for i in model.I)
55 )
56 def linearize_energy_yield_rule(m, k):
57     return m.E_k[k] <= params['y_k'][k-1]*m.K_installed + params['M
58     ']*(1-m.z_k[k])
59 model.LinearizeAnnualEnergyYield = Constraint(model.K, rule=
60     linearize_energy_yield_rule)
61
62 # Solver
63 solver = SolverFactory('glpk')
64 results = solver.solve(model, tee=True)
65
66 print("Optimal Capex:", model.MinimizeCapex())
```

```
65 print("Panel counts:", [model.n_i[i].value for i in model.I])
66 print("Selected config:", [model.z_k[k].value for k in model.K])
67 print("Installed capacity (kWp):", model.K_installed())
68 print("Annual energy (kWh):", [model.E_k[k].value for k in model.K
       ])
```

# E  COORDINATOR ROUTING TRACE EXAMPLES

This appendix provides five representative coordinator routing traces to demonstrate that the proposed Fully Connected Coordinator performs interpretable, auditable, and debugging-oriented agent assignment rather than opaque heuristic switching. Each routing action records: (i) the detected trigger signal, (ii) the natural-language reasoning trace, (iii) the actionable strategy, (iv) the selected agent, (v) the estimated internal success likelihood, and (vi) the execution outcome. These traces exemplify how the system performs iterative diagnosis, refinement, and verification toward feasible engineering solutions.

---

**Routing Trace Example 1 (Real Extract)**

**Trigger:** Import failure during Pyomo model construction.

**Routing reasoning:**
```
trigger_signal:  ImportError:  missing attribute 'suffixes' from
'pyomo.environ'
target_agent:  modeler
strategy:  Remove invalid import and update Pyomo import statements
priority:  high
estimated_success_rate:  0.40
should_continue:  true
reasoning:  The failure originates from an incorrect import.
This is a structural modeling issue rather than conceptual or
optimization-phase related; the Modeler agent is responsible for
code-level construction.
```

**Debug log:**
```
Record debugging attempt:  modeler - ImportError - failure
Record debugging attempt:  modeler - success - success
```

---

**Routing Trace Example 2 (Structural Feasibility Refinement)**

**Trigger:** Detected non-linear structure after initial formulation.

**Routing reasoning:**
```
trigger_signal:  Model violates linear-programming structure due to
non-linear SoC constraint
target_agent:  analyzer
strategy:  Re-inspect SoC constraint formulation and propose an
LP-compatible alternative
priority:  medium
estimated_success_rate:  0.55
should_continue:  true
reasoning:  The solver reports nonlinearity arising from a division
term in SoC dynamics.  This is a formulation correctness issue that
requires a domain-level reasoning pass before code regeneration.
```

**Debug log:**
```
Record debugging attempt:  analyzer - StructuralNonlinearity -
success
```

## Routing Trace Example 3 (Semantic Interpretation Correction)

**Trigger:** Solver returned infeasible status for bus-level nodal balance.

**Routing reasoning:**
```
trigger_signal:  Infeasibility:  Bus-2 nodal balance violated
across scenarios
target_agent:  verifier
strategy:  Locate imbalance sources, inspect demand-to-bus mapping
semantics
priority:  medium
estimated_success_rate:  0.63
should_continue:  true
reasoning:  The variables d_j represent served demand but were not
properly mapped to the bus topology.  This indicates semantic--not
syntactic--misalignment.
```

**Debug log:**
```
Record debugging attempt:  verifier - InfeasibilityMismatch -
failure
```

## Routing Trace Example 4 (Directed Model Refinement)

**Trigger:** Semantic inconsistency confirmed, directed fix needed.

**Routing reasoning:**
```
trigger_signal:  Verifier confirmed mismatch in demand-to-bus
linkage
target_agent:  modeler
strategy:  Rewrite nodal balance equations using explicit
bus-indexed demand sets
priority:  high
estimated_success_rate:  0.72
should_continue:  true
reasoning:  The root cause is localizable and fixable at the model
construction layer; no conceptual redesign is required.
```

**Debug log:**
```
Record debugging attempt:  modeler - ConstraintReformulation -
success
```

## Routing Trace Example 5 (Final Validation and Commitment)

**Trigger:** Feasible solution obtained; final validation pending.

**Routing reasoning:**
```
trigger_signal:  Feasible solution obtained but pending
specification-level validation
target_agent:  solution_verifier
```

```
strategy:  Validate scenario robustness, SoC boundary satisfaction,
and objective semantics
priority:  high
estimated_success_rate:  0.89
should_continue:  false
reasoning:  All constraints are satisfied; a final conformance
check is required before commitment.

Debug log:
Record debugging attempt:  solution_verifier – FinalValidation –
success
```

These five examples collectively demonstrate that the proposed Fully Connected Coordinator supports multi-step diagnosis and refinement, records interpretable decision rationales, and provides actionable, traceable debugging history rather than opaque routing behavior.

## F  BASELINE FAIRNESS SUMMARY

To ensure transparent and fair comparison across agent frameworks, we summarize the adaptation policies used for all baselines in the EngiAgent evaluation:

- **Unified solver access without enhancing baseline capabilities.** All baselines are granted access to the same numerical solver used by EngiAgent. We do not provide additional templates, verification logic, or domain-specific optimizations. Thus, performance differences arise from reasoning and modeling quality, not from tooling advantages.

- **Minimal enabling fixes only.** When a baseline fails due to syntax errors, unbound variables, or missing numeric values, we apply only the minimum fixes necessary to make it executable. These corrections do not alter its modeling logic or constraints and therefore do not improve its solution quality.

- **Original domain prompts are preserved.** Domain hints are part of each method's intended design. Forcing a shared prompt would alter behavior and artificially weaken baselines. We therefore preserve original prompting strategies and only unify solver access.

- **Transparent compute budget reporting instead of forced equality.** Since agent architectures differ substantially, forcing unified budgets would unfairly penalize some methods. We therefore run all baselines using their recommended configurations and report runtime, token usage, and cost in Sec. 6.4 to enable a transparent cost–performance audit.

## G  FRAMEWORK EXTENSIBILITY AND CROSS-TOOL CAPABILITY

This appendix presents two additional benchmark tasks (P7 and P40) to illustrate that EngiAgent is not restricted to Pyomo-based modeling nor dependent on optimization pipelines or solver-specific behavior. While Pyomo is used in part of the benchmark as a unified feasibility and solver interface, it is not required for model construction; the feasibility-driven coordination mechanism remains unchanged. As shown in Section G.1, EngiAgent operates in the same way on ML–based tasks that treat optimization only as a supporting tool (P7), and as demonstrated in Section G.2, it also handles tasks that involve no optimization or modeling stack at all (P40). These results serve as initial demonstrations that only lightweight adjustments to the Modeler/Analyzer prompts and minor changes to solver configuration and result extraction are sufficient to support heterogeneous modeling paradigms, without modifying the core architecture.

### G.1  ML REGRESSION WITH MILP FEATURE SEARCH

This task builds a regression forecasting model evaluated using Mean Absolute Percentage Error (MAPE). The regression form is constructed through machine-learning numerical estimation, and a

mixed-integer linear program (MILP) is used solely for feature selection. Thus, optimization acts only as a supporting mechanism for exploring model configurations, rather than as the representation of domain logic. Pyomo is used only as a generic solver wrapper; the same formulation can alternatively be implemented via OR-Tools, JuMP, or direct solver APIs without affecting agent behavior or feasibility coordination.

**Core implementation excerpt (ML + MILP; solver-agnostic):**

```
1  # ML regression form (constructed by ML semantics, not symbolic modeling)
2  model.beta_0 = Var(domain=Reals)
3  model.beta_j = Var(range(k), domain=Reals)
4  model.delta_j = Var(range(k), domain=Binary)   # Feature selection
       decision
5
6  # Aux term for product beta_j * delta_j (linearized MILP, tool-agnostic)
7  model.beta_j_delta = Var(range(k), range(n), domain=Reals)
8  for j,t in product(range(k), range(n)):
9      model.beta_j_delta[j,t] <=  M * model.delta_j[j]
10     model.beta_j_delta[j,t] >= -M * model.delta_j[j]
11     model.beta_j_delta[j,t] >=  model.beta_j[j] - M*(1-model.delta_j[j])
12     model.beta_j_delta[j,t] <=  model.beta_j[j] + M*(1-model.delta_j[j])
13
14  # Regression prediction used directly by Analyzer & Verifier
15  def pred_rule(m, t):
16      return m.y_hat[t] == m.beta_0 + sum(
17          m.beta_j_delta[j,t] * X[j,t] for j in range(k)
18      )
19  model.pred = Constraint(range(n), rule=pred_rule)
20
21  # MAPE objective (supports ML evaluation, not domain construction)
22  model.obj = Objective(
23      expr=(100/n)*sum(model.abs_error[t]/Y[t] for t in range(n))
24          + lambda_cost*sum(model.delta_j[j] for j in range(k))
25  )
26
27  # Any MILP solver may be used; Pyomo only wraps the call
28  results = SolverFactory("cbc").solve(model)
```

### G.2   PURE ML FORECASTING WITHOUT OPTIMIZATION SOLVERS

This task performs forecasting using numerical machine-learning routines, without constructing an explicit optimization model via modeling languages, invoking a solver, or relying on modeling frameworks. Feasibility is established entirely through statistical and data-consistency checks conducted by EngiAgent's Verifier and handled through the same error-routing mechanism.

**Core implementation excerpt (no solver, no modeling stack):**

```
1  # Closed-form OLS estimation (pure numerical ML; no solver backend)
2  X = np.column_stack([np.ones(n), X1, X2, X3])
3  theta = np.linalg.inv(X.T @ X) @ X.T @ Y
4
5  # Prediction computed entirely by ML numerical routines
6  y_hat = X @ theta
7
8  # Feasibility validation (handled by Verifier, independent of solvers)
9  if np.any(np.isnan(theta)) or np.any(np.isinf(theta)):
10     raise ValueError("Infeasible ML estimation")
```

### G.3   SUMMARY AND FUTURE DIRECTIONS

These extended cases show that EngiAgent supports both solver-dependent ML optimization modeling (e.g., feature search) and solver-free ML tasks (e.g., closed-form OLS estimation) without depending on Pyomo, symbolic modeling frameworks, or specific solver backends. The core architecture

remains unchanged; only tool-level interactions require minor adjustments in prompts and output handling.

This tool-agnostic coordination suggests a natural pathway toward broader engineering domains, such as PDE-based simulation, stochastic system modeling, and hybrid data–physics applications, where specialized estimators or solvers can be introduced while maintaining the same feasibility-centered agent workflow.

