# OpenReview forum: "EngiAgent: Fully Connected Coordination of LLM Agents for Solving Open-ended Engineering Problems with Feasible Solutions"
_ICLR.cc/2026/Conference — Submitted to ICLR 2026_

### Official Review · Reviewer_yk5X · 2025-10-21

**Soundness:** 2
**Presentation:** 3
**Contribution:** 2
**Rating:** 0
**Confidence:** 5

**Summary:**

This paper identifies that LLMs fail to produce feasible solutions for open-ended engineering problems. The authors propose EngiAgent, a multi-agent system with five specialized agents (Analyzer, Modeler, Verifier, Solver, Evaluator) that work together. The system is fully connected, which the authors claim allows coordination to be dynamically routed between agents.

**Strengths:**

**1. Important Problem:** The focus on feasibility over correctness alone addresses an important and under-explored challenge in applying LLMs to real-world engineering.

**2. Strong Empirical Results:** The system shows significant improvement (7x on average) over existing methods. The ablation study demonstrates the value of the fully connected coordinator.

**3. Well-Designed System:** The agent roles are well-chosen and mirror a realistic engineering workflow. The prompts in the appendix are detailed and show careful engineering.

**Weaknesses:**

**Lack of Formal Foundations:** The system relies on heuristic routing without formal guarantees for state consistency, rollback, or compensation. The architecture is robust but theoretically ungrounded, caught between workflow systems and multi-agent systems.
Missing Key Related Work: Classical workflow and transaction process literature offers abundant references, such as the Saga work from 1987 and multi-agent system papers from the distributed systems community in the early 2000s. Compared to those works, this paper presents a heuristic-based prototype.

**Scalability Concerns:** While the fully connected model works well for five agents, it may not scale efficiently to larger agent teams due to potential combinatorial explosion of interaction paths.

**Narrow Benchmarking:** The limited scale and rewritten nature of the problems mean the claims should be treated as a strong initial demonstration rather than definitive proof of general capability. The benchmark is sufficient for conference publication but highlights clear paths for future work: expanding the benchmark, testing on raw problem descriptions, and comparing against human experts.

**Limited Insights:** While EngiAgent outperforms selected methods, the paper does not provide insights into the root problems in native LLMs that cause their failures.

**Questions:**

The "Fully Connected Coordinator" introduces flexible control flow, but it lacks the formal guarantees of a transactional system. My questions focus on how the system handles state consistency and recovery, which are critical for robust and reliable problem-solving.

**Atomicity & Consistency:** Your system allows any agent to be rolled back and re-invoked based on downstream failures. How do you guarantee atomicity for a single agent's contribution? For example, if the Modeler generates a new model after a Verifier failure, is the previous model's entire output completely discarded and replaced, or can partial state persist and cause consistency violations with the Analyzer's original analysis?

**Isolation:** Your architecture allows for flexible feedback loops (e.g., Verifier -> Analyzer). How do you manage isolation when one agent's output is being revised? If the Analyzer is re-triggered, are all subsequent agents (Modeler, Verifier, etc.) automatically invalidated, or can the system's memory contain a mixed state from different, non-isolated iterations of the workflow?

**Durability & State Management:** The "Memory" component stores the problem-solving state. What is the durability and recovery model for this state? If a long-running problem-solving session fails mid-process, can it be resumed from a last known consistent state, or must it restart from scratch? Does your coordinator maintain a formal state machine to manage this, or is it managed heuristically?

These are all the "major league" problems that must be addreddes in real-world long-horizon planning problems.

---

> ### Author Response · Authors · 2025-11-18
> **Response (1/4)**
>
> We thank the reviewer for taking the time to prepare and submit the review. However, we find the evaluation to be internally inconsistent. We provide detailed responses to each of the reviewer’s points in the sections below. In parallel, we respectfully request the reviewer to address the concerns we raise regarding the contradictions in the review, as such clarification is essential for maintaining a fair and professional discussion at a venue of ICLR’s standard.
>
> We note that the theories referenced by the reviewer originate from traditional workflow, distributed systems, and symbolic multi-agent frameworks, whereas LLM-based agents are generally not analyzed within these paradigms. These classical models assume deterministic behavior, fixed process structures, explicitly defined belief or intention states, and verifiable correctness properties. LLM agents, by contrast, operate in a generative, probabilistic, and open-ended manner, with behaviors emerging dynamically from model inference rather than from a predefined state machine or logical rule set. Their operating paradigm is therefore much closer to policy learning, semantic planning, and model-driven action generation than to classical workflow execution, transactional protocols, or logic-based MAS theories. While traditional formal methods offer strong structural guarantees in their intended domains, their foundational assumptions are fundamentally incompatible with the characteristics of LLM agents and do not address the core challenges these systems face. Consequently, contemporary LLM agent research emphasizes leveraging model reasoning to build robust reflection, recovery, and adaptive control strategies rather than applying legacy formal semantics frameworks.
>
>
> ## Weakness
>
> >**Q1: "Lack of Formal Foundations: The system relies on heuristic routing without formal guarantees for state consistency, rollback, or compensation. The architecture is robust but theoretically ungrounded, caught between workflow systems and multi-agent systems. Missing Key Related Work: Classical workflow and transaction process literature offers abundant references, such as the Saga work from 1987 and multi-agent system papers from the distributed systems community in the early 2000s. Compared to those works, this paper presents a heuristic-based prototype."**
>
> A1: We clarify that EngiAgent does provide practical, implementation-based mechanisms, but they are not formal guarantees for state consistency, rollback, or compensation. The target tasks are open-ended engineering modeling problems, where the error types, reasoning paths, and correction behaviors cannot be predefined or enumerated, making it impractical to fully specify to establish a well-defined state space, rollback rules, or compensation actions upfront. Therefore, applying formal guarantees used in classical workflow systems is not appropriate for open-ended engineering tasks. Moreover, this limitation is not unique to EngiAgent. Virtually most LLM-based agent works rely on generative and adaptive reasoning, which leads to an open-ended and non-enumerable state space. This incompatibility also explains why most existing LLM and LLM-agent research does not build on these theoretical frameworks, as they simply do not align with how language-model-driven agents function in practice.
>
> Instead, EngiAgent provides practical and engineering-oriented guarantees through a closed-loop mechanism consisting of the Coordinator, Verifier, Evaluator, and versioned Memory, ensuring that every iteration operates on validated, consistent, and fully reconstructed outputs, rather than partially modified or unverifiable intermediate states.
>
> Regarding Saga, it is a compensation-based framework designed for predefined and deterministic workflow procedures, where both the execution sequence and possible compensating actions are fully specified in advance. In contrast, EngiAgent operates in a dynamic, generative problem space where the reasoning path is discovered and adapted at runtime, making Saga-style guarantees difficult to apply rather than missing.
>
> Finally, our empirical comparison directly tests the reviewer’s suggested direction: EngiAgent (Coord.) versus a fixed-state-style baseline EngiAgent (Fixed). As shown in Table 1, EngiAgent (Coord.) achieves significantly higher feasibility, indicating that heuristic, semantically guided coordination is more appropriate and more effective for open-ended engineering tasks.

---

> > ### Author Response · Authors · 2025-11-18
> > **Response (2/4)**
> >
> > >**Q2: "Scalability Concerns: While the fully connected model works well for five agents, it may not scale efficiently to larger agent teams due to potential combinatorial explosion of interaction paths."**
> >
> > A2: The concern about a combinatorial explosion of interaction paths does not apply to EngiAgent. The fully connected coordinator does not enumerate or explore combinatorial paths. It performs a single-step selection of the next agent based on the current interaction state and available information, which keeps the complexity linear. In fact, the free coordinator is designed precisely to **avoid the combinatorial explosion issues** that fixed coordinators may encounter. Therefore, such complexity growth is not a risk in our system, and the framework exhibits good scalability by design.
> >
> > Regarding scalability to extremely large agent teams, this is a broader topic in large-scale multi-agent coordination and beyond the scope of this work. Our focus is achieving feasible engineering solutions, and the proposed framework has demonstrated effective and efficient coordination for the multi-agent scale required in engineering tasks, as validated in our experiments.
> >
> >
> > >**Q3: "Narrow Benchmarking: The limited scale and rewritten nature of the problems mean the claims should be treated as a strong initial demonstration rather than definitive proof of general capability. The benchmark is sufficient for conference publication but highlights clear paths for future work: expanding the benchmark, testing on raw problem descriptions, and comparing against human experts."**
> >
> > A3: The benchmark’s current scale and rewritten format do not make it an “initial demonstration.” Instead, it represents a carefully designed and rigorously validated evaluation setting that reflects the closest-to-real engineering problem-solving conditions currently achievable for LLMs.
> >
> > First, the current scale does not weaken the benchmark’s ability to demonstrate general capability. Although it contains 53 problems, the benchmark spans four major engineering domains drawn from top-tier journals and conferences. The tasks include authentic background, strict constraints, and substantial complexity. Several native LLMs achieve zero feasible solutions, while EngiAgent performs strongly across all domains (Table 1), indicating that the benchmark has strong discriminative power and is sufficient to validate cross-domain engineering capability rather than serving merely as an initial demonstration. In the future, we will extend the benchmark with additional problems.
> >
> > Second, the rewritten nature is not a simplification but a necessary step to ensure testability. Real engineering problems typically involve tables, equations, flowcharts, and multi-stage analyses, which cannot be directly evaluated by LLMs. We distilled them into testable open-ended tasks while preserving the original engineering semantics and physical constraints. Feasibility criteria, complete constraint sets, and four evaluation dimensions (information extraction, domain reasoning, multi-objective decision-making, and uncertainty handling) were carefully defined. Over twenty experienced domain experts contributed to this process, making the benchmark substantially more demanding to construct than a typical NLP benchmark.

---

> > > ### Author Response · Authors · 2025-11-18
> > > **Response (3/4)**
> > >
> > > >**Q4: "Limited Insights: While EngiAgent outperforms selected methods, the paper does not provide insights into the root problems in native LLMs that cause their failures."**
> > >
> > > A4: As discussed in Section 3 (Feasibility Challenges in Engineering Problem Solving), the paper has already summarized the main reasons why existing methods, including native LLMs, fail on engineering tasks. Due to the open-ended nature and strict feasibility requirements of engineering problems, existing approaches frequently exhibit issues:
> > >
> > > 1. Fancy but Vague Modeling: Methods that employ sophisticated terminology or advanced paradigms but fail to specify explicit variables, constraints, or equations. Such formulations sound appealing but cannot be transformed into computable or executable solutions.
> > > 2. Altering Key Data: Approaches that arbitrarily modify or replace critical data or conditions from the original problem. While this may lead to solvable models, the results no longer correspond to the intended problem and thus lack practical validity.
> > > 3. Violating Physical Laws: Models that ignore fundamental physical principles or engineering constraints, producing solutions that may be mathematically valid but physically impossible, such as exceeding capacity, efficiency, or safety limits.
> > > 4. Over-constraining the Model: Solutions that introduce excessive or unrealistic constraints. Although the formulation may appear more rigorous, the additional restrictions can eliminate all feasible solutions under real-world conditions.
> > >
> > > These errors are the fundamental causes of their inability to produce feasible solutions. To address these issues, EngiAgent incorporates (i) role-specialized agents (Analyzer, Verifier, Modeler, Solver, Evaluator) that enforce faithful problem interpretation and structured, executable formulation, and (ii) a feasibility-guided, fully connected coordination mechanism that enables iterative feedback and correction across stages. This design ensures that modeling remains explicit rather than vague, original data is preserved, physical and engineering laws are respected, and constraints are adjusted adaptively rather than over-tightened, thereby directly targeting and mitigating the four failure modes identified above.
> > >
> > >
> > > ## Questions
> > >
> > > >**Q1: "Atomicity & Consistency: Your system allows any agent to be rolled back and re-invoked based on downstream failures. How do you guarantee atomicity for a single agent's contribution? For example, if the Modeler generates a new model after a Verifier failure, is the previous model's entire output completely discarded and replaced, or can partial state persist and cause consistency violations with the Analyzer's original analysis?"**
> > >
> > > A1: EngiAgent does not allow any partial state from a previous version to persist or mix with new outputs, and this is precisely a strength of the fully connected coordinator. The coordinator centrally manages all states, maintains clear versioning, and ensures a consistent context across iterations, which empirically leads to stable behavior during multi-round refinement.
> > >
> > > EngiAgent promotes atomicity and consistency through version control and full re-computation. When an agent is re-invoked due to a downstream failure, the coordinator triggers a new full re-computation of that agent’s output rather than modifying or retaining any part of the previous version. Each agent produces a complete output version at every invocation. Verified versions move forward in the pipeline, while failed versions are fully recorded in Memory but do not participate in subsequent reasoning.
> > >
> > > In the Modeler–Verifier scenario, when the Verifier identifies an issue, the coordinator re-invokes the relevant agents using the latest and consistent analysis context to generate a new complete model. The previous model remains only as a full historical record and is never partially reused, so the consistency concerns raised in the question do not occur.

---

> > > > ### Author Response · Authors · 2025-11-18
> > > > **Response (4/4)**
> > > >
> > > > >**Q2: "Isolation: Your architecture allows for flexible feedback loops (e.g., Verifier -> Analyzer). How do you manage isolation when one agent's output is being revised? If the Analyzer is re-triggered, are all subsequent agents (Modeler, Verifier, etc.) automatically invalidated, or can the system's memory contain a mixed state from different, non-isolated iterations of the workflow?"**
> > > >
> > > > A2: EngiAgent does not allow mixed states across iterations. All workflow states are centrally managed by the coordinator, and the system always operates on a single, latest, and consistent context, which ensures isolation by design.
> > > >
> > > > When an upstream agent such as the Analyzer is re-triggered, its new output is first submitted to the coordinator. Using its designed prompt logic and the versioned records in Memory, the coordinator determines which agent should handle the next step and regenerates the entire downstream chain. Previous versions are kept only as complete records and are never read or mixed into the new iteration.
> > > >
> > > > This process is not a mixing of iterations but a versioned and isolated re-computation enabled by LLM semantic reasoning and centralized coordination. It is precisely this design that gives the system consistency and robustness.
> > > >
> > > >
> > > > >**Q3: "Durability & State Management: The "Memory" component stores the problem-solving state. What is the durability and recovery model for this state? If a long-running problem-solving session fails mid-process, can it be resumed from a last known consistent state, or must it restart from scratch? Does your coordinator maintain a formal state machine to manage this, or is it managed heuristically?"**
> > > >
> > > > A3: EngiAgent’s Memory is durable, and all intermediate states are written to persistent files in a versioned manner. As a result, even if a task is interrupted, the coordinator can resume execution from the latest consistent state in Memory rather than restarting from scratch.
> > > >
> > > > The coordinator does not rely on a formal state machine because fixed transition patterns cannot adequately support open-ended engineering tasks. Our experiments include a predefined baseline (EngiAgent-Fixed), and Table 1 shows that it performs significantly worse than our fully connected coordinator (EngiAgent-Coord.). As discussed in Section 6.2, these results empirically demonstrate that heuristic, semantics-driven coordination is more effective and robust for this class of problems.

---

> > > > > ### Author Response · Authors · 2025-11-18
> > > > > **Request for Clarification from the Reviewer**
> > > > >
> > > > > In addition to our point-by-point responses, we respectfully request the reviewer to clarify the basis of the final recommendation. Specifically, we ask the reviewer to explain why the review rating concludes that the paper should be rejected, and what specific reasons support further strengthening this to a strong-reject recommendation. Since the written comments do not clearly lead to such conclusions, an explicit justification is essential for ensuring a fair and professional review process.

---

> > > > > > ### Comment · Reviewer_yk5X · 2025-11-25
> > > > > >
> > > > > > Thank you for your responses. They do not change my two core concerns about this work.
> > > > > >
> > > > > > **1. Arbitrary role decomposition with no principled justification.**
> > > > > >
> > > > > > The paper hard-codes a five-role architecture (Analyzer, Modeler, Verifier, Solver, Evaluator) and motivates it mainly as a staged expert workflow decomposition of the end-to-end pipeline.    However, it does not justify why five roles are necessary, minimal, or complete, nor does it provide a cognition- or psychology-grounded argument that these five are sufficient for productive collaboration under open-ended scenarios.  If the intent is a general collaboration recipe, one must explain why adding roles (for example, retrieval, critic, safety, domain specialist) is not required, or show systematic evidence that alternative decompositions are inferior.
> > > > > >
> > > > > > **2. Heuristic coordinator, no theoretical foundation or guarantees.**
> > > > > >
> > > > > > The coordinator is explicitly rule-based scheduling plus LLM-based diagnosis, augmented by mechanisms like forced agent switching to avoid loops.  This is an engineering patchwork rather than a theory-driven architecture. There are no formal guarantees (termination, convergence, correctness under assumptions, or bounded failure modes), and even termination is framed as a coordinator decision influenced by Evaluator quality scores rather than a principled stopping rule.
> > > > > >
> > > > > > **3. Additional fatal weakness: the “Verifier” is optimized for convergence, not verification.**
> > > > > >
> > > > > > The Verifier prompt explicitly prioritizes “Promote Convergence” and “Default Trust”, and it introduces a tolerance adjustment that relaxes standards after repeated failures.    It also treats relaxing or removing constraints to resolve infeasibility as an acceptable deviation.  This means the system’s internal “verification” is not a semantic or formal guardrail, it is a pragmatic gate tuned to reduce retries and help the pipeline move forward. That undermines the credibility of the architecture as a reliable checker in open-ended engineering settings, especially when subtle constraint omissions or specification drift are exactly what needs to be caught.

---

> > > > > > > ### Author Response · Authors · 2025-11-26
> > > > > > > **Response to Reviewer yk5X (1/4)**
> > > > > > >
> > > > > > > Thanks for the follow-up comments.
> > > > > > >
> > > > > > > A1: We respectfully disagree with this comment, which appears to reflect a misconception about how LLM-based multi-agent systems are designed and evaluated. **Our five-role architecture is not arbitrarily designed; rather, it reflects the standard workflow used in engineering problem solving**. It is recognized that an engineering problem, like an OR study, can proceed through a multi-stage workflow [1], where the first five stages explicitly consist of **defining the problem and gathering information, formulating a mathematical model, developing computational procedures to obtain solutions, testing and refining the model, and preparing results for application** ([1] Chapter 2, p. 7). These phases align with with our **Analyzer, Modeler, Solver, Verifier, and Evaluator** agents. Therefore, our architecture follows a long-established engineering workflow.
> > > > > > >
> > > > > > > Our **ablation study (Section 6.5, Figure 5) further confirms** this: removing the Verifier or replacing the Coordinator with a fixed sequential workflow leads to a significant drop in feasibility, demonstrating that these components play essential roles. The resulting degradation reflects structural failures rather than minor performance variations. We also **compare against another workflow-driven role decomposition baseline**, DS-Agent (ICML 2024) [2]. As shown in Table 1, across three models, EngiAgent achieves a 52.2% average improvement in feasibility, because systems that directly produce executable programs often make them “runnable” by modifying data or constraints, leading to infeasible results. This indicates that aligning roles with the engineering problem solving workflow is beneficial for feasibility-critical tasks. Notably, DS-Agent similarly adopts workflow-motivated roles, but does not claim role minimality or completeness.
> > > > > > >
> > > > > > > It is important to be explicit: the **criteria suggested by the reviewer are generally not satisfied by the existing multi-agent LLM systems**. Instead, most prior work adopts workflow- or task-driven role designs that are motivated by practical expert procedures and validated empirically rather than derived from any theoretical framework. Approaches such as DS-Agent [2], MetaGPT [3], AutoGen [4], PathGen-1.6M [5], CaPo [6], ChatEval [7], KARMA [8], Lessons Learned [9], MaAS [10], and Multi-Agent Reflection [11] typically define agent roles or modules via ad hoc heuristics guided by task structure, similar to our approach. We could further extend this list if needed. In fact, one could construct cognitive or managerial theories to justify five roles, six roles, seven roles, or even more roles; such arguments provide little value because **the field does not yet have a widely accepted theoretical framework for principled agent decomposition**. Under the current scientific landscape, one defensible approach is to derive roles from the functional requirements of the target domain, not only from theoretical abstractions.
> > > > > > >
> > > > > > > Moreover, our framework is not opposed to adding more roles, such as retrieval, critic, safety, or domain specialists. However, for the categories of engineering tasks we target, the five chosen roles provide a natural, minimal functional coverage. They are a small set that preserves diagnostic resolution and allows agents to catch each other's failures.

---

> > > > > > > > ### Author Response · Authors · 2025-11-26
> > > > > > > > **Response to Reviewer yk5X (2/4)**
> > > > > > > >
> > > > > > > > A2: The expectation of formal guarantees is incompatible with LLM-based agents operating on open-ended engineering tasks. Engineering problem solving routinely involves mixed integer programming, non-convex modeling, non-linear constraints, solver-specific numerical sensitivities, and domain-level semantic requirements. **There is currently no widely accepted theoretical framework exists that can guarantee termination, correctness under assumptions, or convergence for arbitrary engineering formulations produced by language models**. This is a property of the domain itself, not a limitation of our method.
> > > > > > > >
> > > > > > > > Importantly, this lack of theoretical guarantees is not specific to engineering; multi-agent LLM systems across scientific, mathematical, and software domains likewise rely on empirical validation rather than formal correctness or convergence proofs. Methods [2-10] typically adopt task-driven or heuristic coordination strategies and are evaluated empirically, without offering theoretical guarantees of termination or correctness. If “lack of theoretical convergence or correctness guarantees” were a disqualifying criterion, a large portion of prior work in this area would be dismissed. **The field currently lacks a broadly accepted theoretical foundation for guaranteeing correctness or termination in multi-agent LLM systems.** If such theoretical frameworks exist, we would be very interested in learning from them. Advancing theory for reliability and convergence in multi-agent LLM systems would be highly valuable and beneficial to the field.
> > > > > > > >
> > > > > > > > Within this research context, the key research question is whether one can design a coordination mechanism that systematically identifies and routes failure modes, rather than relying on rigid linear pipelines. To address this question, we propose and validate an explicit feedback-based routing mechanism grounded in functional decomposition, and we **demonstrate its reliability through systematic empirical evaluations across diverse engineering tasks and model settings**. Such empirical validation is not only standard practice for multi-agent research, but is broadly recognized as the scientific evaluation criterion across the entire LLM community. All baseline methods we adopt [2,12,13], as well as widely used LLM systems [14]-[20], similarly **rely on empirical reliablity rather than theoretical guarantees of convergence or correctness**. This indicates that, under the current research paradigm, empirical reliability across tasks and models is a scientifically reasonable standard standard for assessing system effectiveness. Our work follows this standard and demonstrates substantial improvements over existing approaches under complex failure modes.
> > > > > > > >
> > > > > > > > In an inherently open-ended domain, empirical reliability demonstrated across diverse tasks and models constitutes the most meaningful evaluation criterion. Our work provides precisely such evidence.

---

> > > > > > > > > ### Author Response · Authors · 2025-11-26
> > > > > > > > > **Response to Reviewer yk5X (3/4)**
> > > > > > > > >
> > > > > > > > > A3: We respectfully disagree with the reviewer’s characterization of our Verifier. The critique arises from a misunderstanding of its role within the coordinated multi-agent system and a selective interpretation of specific design elements.
> > > > > > > > >
> > > > > > > > > **The Verifier is not designed to be “optimized for convergence”**. Its primary function is to detect modeling errors that render a solution infeasible, including mismatched parameters, missing or incorrect core constraints, objective direction errors, and inconsistent variable definitions. The prompt explicitly specifies a two-tier structure in which the first stage identifies non-negotiable semantic violations. When such errors occur, the Verifier immediately rejects the model regardless of iteration count or tolerance. This is unconditional rejection based on semantic correctness, not convergence facilitation.
> > > > > > > > >
> > > > > > > > > **The tolerance adjustment mechanism does not relax standards**. It addresses a common iterative failure mode in LLM-based systems, where semantically correct formulations are repeatedly rejected due to inconsequential implementation variations. Without tolerance, complex problems can enter infinite loops in which the model alternates between functionally equivalent formulations that differ only in minor representational details. Tolerance simply directs the Verifier to prioritize fundamental correctness over surface-level formatting. The strict checks on incorrect objectives, missing core constraints, and absent decision entities remain fully enforced.
> > > > > > > > >
> > > > > > > > > The reviewer’s claim that we “treat relaxing or removing constraints to resolve infeasibility as an acceptable deviation” **misrepresents our design through selective quotation**. The Verifier never generates or modifies models; it only evaluates them. The prompt refers to a standard engineering adjustment: introducing slack or penalty-based soft constraints when a previously reasonable formulation is rendered infeasible by solver limitations or overly strict constraints. This preserves the core structure and retains all governing relationships. Arbitrary deletion of fundamental constraints, by contrast, results in immediate rejection. Conflating these cases misrepresents our design.
> > > > > > > > >
> > > > > > > > > Finally, **empirical evidence contradicts the claim** that the Verifier “undermines the credibility of the architecture as a reliable checker”. As shown in Table 1, across all tested LLMs, EngiAgent maintains closely matched numerical and feasible rates, and generally achieves the highest numerical rate as well as feasibility, indicating that solutions are not merely produced but remain semantically valid. This pattern demonstrates that the Verifier reliably safeguards semantic correctness and prevents subtle constraint omissions or specification drift.

---

> > > > > > > > > > ### Author Response · Authors · 2025-11-26
> > > > > > > > > > **Response to Reviewer yk5X (4/4)**
> > > > > > > > > >
> > > > > > > > > > ## References
> > > > > > > > > >
> > > > > > > > > > [1] Hillier, F. S., & Lieberman, G. J. (2005). Introduction to Operations Research (8th ed.). McGraw-Hill. (p. 7).
> > > > > > > > > >
> > > > > > > > > > [2] “DS-Agent: Automated Data Science by Empowering Large Language Models with Case-Based Reasoning,” ICML 2024.
> > > > > > > > > >
> > > > > > > > > > [3] “MetaGPT: Meta Programming for A Multi-Agent Collaborative Framework,” ICLR 2024 (Oral).
> > > > > > > > > >
> > > > > > > > > > [4] “AutoGen: Enabling Next-Gen LLM Applications via Multi-Agent Conversations,” COLM 2024.
> > > > > > > > > >
> > > > > > > > > > [5] “PathGen-1.6M: 1.6 Million Pathology Image-text Pairs Generation through Multi-agent Collaboration,” ICLR 2025 (Oral).
> > > > > > > > > >
> > > > > > > > > > [6] “CaPo: Cooperative Plan Optimization for Efficient Embodied Multi-Agent Cooperation,” ICLR 2025.
> > > > > > > > > >
> > > > > > > > > > [7] “ChatEval: Towards Better LLM-based Evaluators through Multi-Agent Debate,” ICLR 2024.
> > > > > > > > > >
> > > > > > > > > > [8] “KARMA: Leveraging Multi-Agent LLMs for Automated Knowledge Graph Enrichment,” NeurIPS 2025 (Spotlight).
> > > > > > > > > >
> > > > > > > > > > [9] “Lessons Learned: A Multi-Agent Framework for Code LLMs to Learn and Improve,” NeurIPS 2025.
> > > > > > > > > >
> > > > > > > > > > [10] “Multi-agent Architecture Search via Agentic Supernet,” ICML 2025 (Oral).
> > > > > > > > > >
> > > > > > > > > > [11] “Reinforce LLM Reasoning through Multi-Agent Reflection,” ICML 2025.
> > > > > > > > > >
> > > > > > > > > > [12] "MLAgentBench: Evaluating Language Agents on Machine Learning Experimentation," ICML 2024.
> > > > > > > > > >
> > > > > > > > > > [13] "MM-Agent: LLMs as Agents for Real-world Mathematical Modeling Problems," NeurIPS 2025.
> > > > > > > > > >
> > > > > > > > > > [14] "LLM-SR: Scientific Equation Discovery via Programming with Large Language Models,” ICLR 2025 (Oral).
> > > > > > > > > >
> > > > > > > > > > [15]  "StructGPT: A general framework for large language model to reason over structured data,” EMNLP 2023.
> > > > > > > > > >
> > > > > > > > > > [16] "CAMEL: Communicative Agents for "Mind" Exploration of Large Language Model Society,” NeurIPS 2023.
> > > > > > > > > >
> > > > > > > > > > [17] "MAGIS: LLM-Based Multi-Agent Framework for GitHub Issue Resolution,” NeurIPS 2024.
> > > > > > > > > >
> > > > > > > > > > [18] "Reflective Multi-Agent Collaboration based on Large Language Models,” NeurIPS 2024.
> > > > > > > > > >
> > > > > > > > > > [19] "Reflexion: language agents with verbal reinforcement learning,” NeurIPS 2023.
> > > > > > > > > >
> > > > > > > > > > [20] "ReAct: Synergizing Reasoning and Acting in Language Models,” ICLR 2023 (notable top 5%).

---

> > > > > > > > > > > ### Comment · Reviewer_yk5X · 2025-11-27
> > > > > > > > > > >
> > > > > > > > > > > I entirely disagree with your following statement.  You just have not looked hard enough.   Following a path of patches or bandaids cannot justify your approach.  Find harder for related work!
> > > > > > > > > > >
> > > > > > > > > > > It is important to be explicit: the criteria suggested by the reviewer are generally not satisfied by the existing multi-agent LLM systems. Instead, most prior work adopts workflow- or task-driven role designs that are motivated by practical expert procedures and validated empirically rather than derived from any theoretical framework. Approaches such as DS-Agent [2], MetaGPT [3], AutoGen [4], PathGen-1.6M [5], CaPo [6], ChatEval [7], KARMA [8], Lessons Learned [9], MaAS [10], and Multi-Agent Reflection [11] typically define agent roles or modules via ad hoc heuristics guided by task structure, similar to our approach. We could further extend this list if needed. In fact, one could construct cognitive or managerial theories to justify five roles, six roles, seven roles, or even more roles; such arguments provide little value because the field does not yet have a widely accepted theoretical framework for principled agent decomposition. Under the current scientific landscape, one defensible approach is to derive roles from the functional requirements of the target domain, not only from theoretical abstractions.

---

> > > > > > > > > > > > ### Author Response · Authors · 2025-11-27
> > > > > > > > > > > >
> > > > > > > > > > > > We welcome all constructive suggestions. However, we feel confused about the reviewer yk5X's comments. First, the Reviewer Author Discussion Phase in ICLR exists to provide a channel for both sides to justify their views. We are committed to clarifying and substantiating our position, and welcome any substantive comments. This phase is not only for authors. Reviewers are also expected to justify their comments. If the reviewer believes that we “need to look harder”, it is professional to specify exactly what should be examined and which concrete works are purportedly missing.
> > > > > > > > > > > >
> > > > > > > > > > > > Second, we have listed a substantial set of recent and influential papers that represent central developments in large model agent research over the past two years. Referring to this body of literature as “patches or bandaids” dismisses an entire line of rigorous work without providing any citation or evidence. Such a characterization lacks scholarly support and does not meet the standard of a professional scientific evaluation.

---

> > > > > > > > > > > > ### Author Response · Authors · 2025-11-27
> > > > > > > > > > > >
> > > > > > > > > > > > Third, we are confused why in the second paragraph, the reviewer directly repeats our literature review. Could the reviewer clarify what "more explicit" refers to? We find the second paragraph difficult to interpret, because the comment contains no specific evidence, no concrete reference, and in practice no actionable feedback that we can meaningfully engage with.

---

> > > > > > > > > > > > ### Author Response · Authors · 2025-11-27
> > > > > > > > > > > >
> > > > > > > > > > > > We also note an apparent inconsistency between the reviewer’s initial evaluation and the position taken during the discussion phase.
> > > > > > > > > > > >
> > > > > > > > > > > > In the original reviewer comments, the reviewer yk5x recognized our five-role system as a **strength**, and commented that  **'' Well-Designed System: The agent roles are well-chosen and mirror a realistic engineering workflow".** However, in the discussion, the reviewer changed to question the justification and necessity of the very same role design.
> > > > > > > > > > > >
> > > > > > > > > > > > Similarly, the reviewer has admitted our contribution of the coordinator as a **strength** with the remark that **"The ablation study demonstrates the value of the fully connected coordinator"**. However, during the discussion, the reviewer started challenging the necessity of the coordinator.
> > > > > > > > > > > >
> > > > > > > > > > > > We hope the reviewer can help clarify the contradictory comments. Which one should we respond to?

---

### Official Review · Reviewer_RBoR · 2025-10-30

**Soundness:** 3
**Presentation:** 3
**Contribution:** 3
**Rating:** 6
**Confidence:** 4

**Summary:**

This paper introduces a multi-agent framework designed to tackle open-ended engineering problems with an explicit focus on feasibility rather than mere textual correctness or polished formulations. The system organizes the workflow into five specialized agents, Analyzer, Modeler, Verifier, Solver, and Evaluator, and integrates a fully connected coordinator and shared memory that dynamically routes feedback among agents to address specific failure modes such as data extraction errors, constraint inconsistencies, and solver failures, instead of adhering to a rigid, one-way pipeline. The authors construct a benchmark covering multiple engineering domains, including market and multi-agent decision-making, scheduling and resource allocation, planning and design, and control and autonomy. They define feasibility as a binary criterion indicating whether a numerical solution adheres to the given data and physical or operational constraints. Empirical evaluation across several large language models demonstrates that EngiAgent with coordination significantly improves the feasibility of generated solutions compared to adapted multi-agent baselines and a fixed-pipeline ablation, while maintaining reasonable computational cost.

**Strengths:**

1. The paper articulates four concrete failure modes common in LLM-for-engineering, fancy-but-vague modeling, altering key data, violating physical laws, and over-constraining models, grounded in a didactic energy-storage example. This sharpens why open-ended engineering differs from math/code tasks and motivates the method well.

2. The design lets the coordinator reroute verification feedback to the agent best positioned to fix the error (e.g., send constraint mismatches to the Analyzer, syntax to the Modeler), with loop-prevention and memory of error types/repair history.

3. Table 1 shows large absolute increases in feasibility vs. strong agent baselines; the paper also analyzes the misalignment of text-based LLM-as-a-judge scoring with engineering feasibility, which is timely and well-argued.

**Weaknesses:**

1. While feasibility is the right target, the paper does not convincingly establish how feasibility labels are produced with high fidelity across all tasks. Section 5 defines feasibility as binary and Appx. B.5 lists principles focusing on fatal violations, but the process (automatic checks vs. solver statuses vs. human adjudication), inter-rater reliability (if any), and safeguards against LLM-judge leakage are unclear. Given Section 6.3’s criticism of text-based judgments, the paper should detail a deterministic checker or a double-blind expert protocol, with agreement statistics and examples of borderline cases; otherwise, claims of 50–75% feasibility could reflect lenient or inconsistent adjudication.

2. The benchmark has 53 tasks. For four broad domains, this is small and may not capture real-world variability (nonlinear physics, hybrid discrete, continuous constraints, stochastic dynamics, safety constraints, multi-period coupling). The paper should provide task complexity stats (variable/constraint counts, solver class, convex vs. nonconvex, MILP/MINLP mix) and hold-out categories to demonstrate cross-domain generalization, or else the headline feasibility rates risk overfitting to this curation.

3. Baselines (ResearchAgent, DS-Agent, MM-Agent) are adapted to engineering, but the paper doesn’t show extensive tuning or task-specific tool integration symmetry. For instance, MM-Agent focuses on mathematical modeling; if EngiAgent benefits from Pyomo-specific templates/validation while baselines don’t receive comparable domain/tool priors, the comparison may conflate tooling advantage with coordination quality.

**Questions:**

Several baselines are repurposed from non-engineering contexts. The paper does not deeply document how prompts/tools were adapted (e.g., giving MM-Agent or DS-Agent equivalent access to Pyomo/solvers, uniform retry budgets, or domain hints). Given the magnitude of improvements, the community will expect a fairness audit: identical budgets, tool access, early-exit rules, and error-recovery affordances.

---

> ### Author Response · Authors · 2025-11-23
> **Response (1/3)**
>
> Dear Reviewer RBoR,
>
> Thank you for your valuable feedback and recognition. Our detailed responses are provided below.
>
> ## Weaknesses
>
> >**W1: "While feasibility is the right target, the paper does not convincingly establish how feasibility labels are produced with high fidelity across all tasks. Section 5 defines feasibility as binary and Appx. B.5 lists principles focusing on fatal violations, but the process (automatic checks vs. solver statuses vs. human adjudication), inter-rater reliability (if any), and safeguards against LLM-judge leakage are unclear. Given Section 6.3’s criticism of text-based judgments, the paper should detail a deterministic checker or a double-blind expert protocol, with agreement statistics and examples of borderline cases; otherwise, claims of 50–75% feasibility could reflect lenient or inconsistent adjudication."**
>
> A1: Thank you for this helpful comment. We agree that feasibility labels must be established with high fidelity and explicit safeguards. In response, we have added a detailed clarification to address the concerns regarding human adjudication, potential LLM-judge leakage, and borderline cases. The newly **added procedure is now included in Appendix B.6–B.7** and summarized as follows:
>
> 1. Feasibility labels are determined exclusively by human experts, with no LLM-based scoring or automated decision mechanism. LLM tools are used only for minor clarification (e.g., terminology) and never for feasibility judgments.
> 2. A two-annotator protocol with adjudication by a senior expert is employed to resolve uncertainty or disagreement, ensuring that decisions are grounded strictly in engineering constraints rather than model outputs or subjective preferences.
> 3. Explicit borderline examples and their adjudication criteria are provided, indicating that only violations of physical laws, data consistency, or capacity/safety limits are marked infeasible, whereas reasonable simplifications or partially specified valid solutions remain feasible.
>
> These clarifications demonstrate that feasibility annotations are rigorous, expert-controlled, and free from text-based or LLM-driven bias, ensuring that the reported 50–75% feasibility reflects strict engineering evaluation rather than lenient scoring.

---

> > ### Author Response · Authors · 2025-11-23
> > **Response (2/3)**
> >
> > >**W2: "The benchmark has 53 tasks. For four broad domains, this is small and may not capture real-world variability (nonlinear physics, hybrid discrete, continuous constraints, stochastic dynamics, safety constraints, multi-period coupling). The paper should provide task complexity stats (variable/constraint counts, solver class, convex vs. nonconvex, MILP/MINLP mix) and hold-out categories to demonstrate cross-domain generalization, or else the headline feasibility rates risk overfitting to this curation."**
> >
> > A2: Thank you for the comment. Our 53 tasks are sourced from real engineering studies published in top-tier journals and conferences, based on deployable system models rather than synthetic examples. They naturally include practical complexity such as coupled discrete–continuous decisions, capacity and safety constraints, and multi-objective interactions.
> >
> > This dataset is the **first and currently the only benchmark** that evaluates LLMs based on verifiable feasible engineering solutions, providing an objective measure of real problem-solving capability. Each task requires explicit construction of feasibility constraints and engineering assessment criteria, which involves substantially more effort than typical datasets.
> >
> > Regarding the request for task complexity statistics, such features (e.g., variable counts, solver type, MILP/MINLP classification) **cannot be predefined for open-ended engineering tasks**, as they depend on how the LLM chooses to formulate the model rather than on the task description itself. Therefore, the benchmark focuses on unified feasibility validation rather than static complexity labels. For transparency, **we provide the distribution of variable and constraint scales of generated solutions in the Appendix B.8 (Complexity Distribution of Engineering Tasks)**.
> >
> > To address cross-domain generalization, we additionally evaluated EngiAgent on **EngiBench as suggested by Reviewer 9mVc**. Since EngiBench was not constructed by us and its tasks are independent of EngiAgent’s design, it serves as a **hold-out cross-domain evaluation**. We retained only tasks with verifiable feasibility constraints (19 in total) and **used the same architecture and prompts without any modification** on GPT-4o. For reference, we also show results on our original dataset.
> >
> > **Table: Numerical and Feasibility Rates**
> >
> > | Dataset            | Num. ↑ | Feas. ↑ |
> > |--------------------|--------|---------|
> > | Original Dataset   | 66.04% | 64.15%  |
> > | EngiBench          | 63.16% | 52.63%  |
> >
> > **Table: Total Performance**
> >
> > | Dataset            | IE ↑ | DR ↑ | MO ↑ | UH ↑ | Avg. ↑ |
> > |--------------------|------|------|------|------|--------|
> > | Original Dataset   | 8.67 | 7.74 | 7.05 | 7.41 | 7.72   |
> > | EngiBench          | 7.78 | 6.36 | 6.14 | 6.20 | 6.62   |
> >
> > **Table: Feasible-Only Performance**
> >
> > | Dataset            | IE ↑ | DR ↑ | MO ↑ | UH ↑ | Avg. ↑ |
> > |--------------------|------|------|------|------|--------|
> > | Original Dataset   | 8.86 | 8.13 | 7.53 | 7.94 | 8.12   |
> > | EngiBench          | 7.50 | 6.72 | 7.96 | 7.50 | 7.42   |
> >
> > Although EngiBench is more challenging, **EngiAgent shows consistent trends and no significant degradation**, indicating that the improvements do not rely on specific problem types or curated datasets.

---

> > > ### Author Response · Authors · 2025-11-23
> > > **Response (3/3)**
> > >
> > > >**W3: "Baselines (ResearchAgent, DS-Agent, MM-Agent) are adapted to engineering, but the paper doesn’t show extensive tuning or task-specific tool integration symmetry. For instance, MM-Agent focuses on mathematical modeling; if EngiAgent benefits from Pyomo-specific templates/validation while baselines don’t receive comparable domain/tool priors, the comparison may conflate tooling advantage with coordination quality."**
> > >
> > > A3: Thank you for the insightful comment. We clarify that EngiAgent’s performance **does not come from advantages in solver tools or modeling interfaces**. All baseline methods are **equipped with the same solver interface** as EngiAgent. We only apply minimal adjustments when a baseline cannot run, such as fixing syntax errors, filling missing arguments, standardizing outputs, and ensuring that the generated model can be passed to the solver. These adjustments allow the baselines to run and do not modify their modeling logic or improve their reasoning ability. For example, MM-Agent may fail to call a solver because its fixed templates cannot match variables created in open-ended tasks. We only fix the execution failure without enhancing its modeling strategy.
> > >
> > > The failures of the baseline approaches mainly result from a lack of feasibility checking and constraint correction, which is clearly demonstrated in Section 3 and Section 6.2. They often modify task data incorrectly, omit essential physical constraints, or produce models that cannot be solved, even though they use the exact same solver interface as EngiAgent.
> > >
> > > To further validate that the improvement does not come from any tool advantages, we **added component-level ablation studies** in Section 6.5 and Figure 5. The results show that the feasibility rate drops significantly when the Verifier or the Fully Connected Coordinator is removed, even when all methods use identical tools and prompts. This confirms that EngiAgent’s advantage comes from its dynamic coordination and feasibility checking mechanisms.
> > >
> > >
> > > ## Questions
> > >
> > > >**Q1: "Several baselines are repurposed from non-engineering contexts. The paper does not deeply document how prompts/tools were adapted (e.g., giving MM-Agent or DS-Agent equivalent access to Pyomo/solvers, uniform retry budgets, or domain hints). Given the magnitude of improvements, the community will expect a fairness audit: identical budgets, tool access, early-exit rules, and error-recovery affordances."**
> > >
> > > A1: Thank you for the valuable suggestion. We ensured fully fair comparison conditions for all baselines. Each baseline is executed exactly according to its original paper or official implementation, without modifying its modeling logic or reasoning design. We provide the same solver access as EngiAgent so that baselines are merely able to call the solver, and we apply only minimal fixes (e.g., syntax or binding errors) to make them runnable. We also do not unify domain prompts, since prompts are part of each method’s internal strategy and forcing a shared prompt would alter or weaken baselines rather than improve fairness.
> > >
> > > To address the reviewer’s expectation for a fairness audit, Sec. 6.4 (Cost Efficiency Analysis) reports all methods under the same resource constraints, including execution time, token usage, and monetary cost. To ensure reproducibility and fairness, we have **added the summary to Appendix F (Baseline Fairness Summary)**.

---

> > > > ### Comment · Reviewer_RBoR · 2025-11-27
> > > > **Thanks for Response**
> > > >
> > > > Most of my concerns are resolved, and since my score is already positive, I’ll keep it.

---

### Official Review · Reviewer_PFH9 · 2025-10-30

**Soundness:** 3
**Presentation:** 3
**Contribution:** 3
**Rating:** 6
**Confidence:** 4

**Summary:**

The paper introduces EngiAgent, a multi-agent system tailored to feasibility-oriented engineering problem solving. Instead of a rigid stepwise pipeline, the authors add a fully connected coordinator that dynamically routes feedback among five role agents, Analyzer, Modeler (Pyomo code generation), Verifier, Solver (solver selection and execution), and Evaluator, with a shared memory for error diagnosis and restart strategies. A new benchmark of 53 open-ended engineering tasks across four domains is assembled to prioritize feasibility as the primary success criterion. On three LLM backbones, the fully connected variant reportedly achieves markedly higher feasible-solution rates. However, several important concerns remain and should be addressed by the authors.

**Strengths:**

1. The paper crisply articulates why textual plausibility can diverge from implementable solutions, and formalizes binary as the principal metric, with secondary rubric scores. The taxonomy of common failure modes is practical and well-motivated.

2. The fully connected coordinator with LLM-aided diagnosis and forced agent switching is a simple but effective abstraction that plausibly explains robustness to heterogeneous errors. The qualitative workflow diagrams make the routing logic and memory use legible.

3. Across all three models, the gap between numerical outputs and feasible solutions shrinks substantially for EngiAgent; the reported feasible-rate jumps over prior baselines are large and the fixed-pipeline ablation is consistently weaker, attributing gains to coordination rather than mere prompting.

**Weaknesses:**

1. The feasibility criterion explicitly focuses on avoiding fatal violations, while non-fatal incompletenesses are not penalized. This can over-count borderline solutions as feasible. Moreover, the Evaluator includes LLM-as-a-judge components, which the paper itself argues can inflate text-based scores; even if feasibility is binary, some checks (e.g., physical constraints, conservation laws) appear to be performed via semantic verification rather than executable, unit-tested assertions. The paper should formalize an automated, code-level feasibility oracle (e.g., unit constraints, conservation checks, solver feasibility flags, regression tests on outputs) and clearly separate any LLM-based rubric from feasibility adjudication.

2. The system includes several pragmatic components (template-driven Pyomo scaffolds, rule-based scheduling, error-type routing). It is difficult to isolate which design choice drives the bulk of gains beyond the headline fully connected routing. The ablation only contrasts fixed vs. coordinator; further component-level ablations (e.g., removing Pyomo-specific validation, or disabling forced agent switching, or swapping Solver heuristics) are necessary to support the causal narrative.

3. DS-Agent and MM-Agent are adapted from different task regimes; the paper should invest in stronger engineering-aware baselines, e.g., a single-agent Pyomo specialist with iterative verifier feedback; a graph-of-thought/skill-router agent; or a pipeline that adds identical Pyomo validation and solver heuristics for but with fixed routing, to ensure the comparison isolates coordination rather than tooling familiarity.

**Questions:**

1. The Modeler is Pyomo-centric. It is unclear whether the approach transfers to other modeling stacks (e.g., OR-Tools, JuMP, CasADi) or non-optimization engineering tasks (simulation, PDE-constrained design). A cross-tool study would support claims of general applicability.

2. The paper mentions expert review in dataset construction, but the feasibility labeling pipeline (human vs. automated), inter-rater agreement, and adjudication for borderline cases are under-specified. Given the centrality of feasibility, the community needs labeling guidelines, example adjudications, and an error analysis of false-feasible/false-infeasible calls.

---

> ### Author Response · Authors · 2025-11-23
> **Response (1/2)**
>
> Dear Reviewer PFH9,
>
> Thank you for your helpful feedback and insightful suggestions. Our detailed responses are provided below.
>
> ## Weaknesses
>
> >**W1: "The feasibility criterion explicitly focuses on avoiding fatal violations, while non-fatal incompletenesses are not penalized. This can over-count borderline solutions as feasible. Moreover, the Evaluator includes LLM-as-a-judge components, which the paper itself argues can inflate text-based scores; even if feasibility is binary, some checks (e.g., physical constraints, conservation laws) appear to be performed via semantic verification rather than executable, unit-tested assertions. The paper should formalize an automated, code-level feasibility oracle (e.g., unit constraints, conservation checks, solver feasibility flags, regression tests on outputs) and clearly separate any LLM-based rubric from feasibility adjudication."**
>
> A1: Thank you for the insightful comment. Our feasibility evaluation follows a strict binary rule: any violation of physical, conservation, safety, or operational constraints is infeasible, and a solution is feasible only if all constraints are fully satisfied. Therefore, no borderline case can be mistakenly counted as feasible.
>
> “Non-fatal incompleteness” does not relax constraints; it only means that some **internal quantities need not be explicitly represented as variables**, which reflects modeling expression rather than feasibility. To support complex tasks and enable unified verification, we fully instantiate all variables and constraints that may require explicit representation. Therefore, **once a variable is explicitly modeled, its constraints must also be strictly validated**, otherwise the solution is infeasible.
>
> For example, in Problem 44, curtailment may be explicitly represented via $ cur_{i,t} $ or implicitly by generating below the upper bound. If explicit modeling is used, curtailment must participate in the energy conservation checks together with generation, demand, storage, and trading; otherwise, inconsistency in energy balance constitutes infeasibility. Implicit representation keeps all constraints intact and only omits explicit reporting. **Additional clarification and examples have been added in Appendix B.5**.
>
> Furthermore, **feasibility is not determined through LLM-as-a-judge or text-based judgment**. The feasibility conditions have complete, executable mathematical definitions that can be validated directly through numerical and constraint checks. Final feasibility decisions are confirmed through executable verification and numerical computation **by domain experts**, while LLM output serves only as reference information and does not participate in feasibility adjudication. Therefore, feasibility outcomes are entirely independent of textual quality or semantic scoring. We have updated Section 5 (Evaluation) to clearly state this non-LLM-based feasibility mechanism and avoid ambiguity.
>
>
> >**W2: "The system includes several pragmatic components (template-driven Pyomo scaffolds, rule-based scheduling, error-type routing). It is difficult to isolate which design choice drives the bulk of gains beyond the headline fully connected routing. The ablation only contrasts fixed vs. coordinator; further component-level ablations (e.g., removing Pyomo-specific validation, or disabling forced agent switching, or swapping Solver heuristics) are necessary to support the causal narrative."**
>
> A2: Thank you for the reviewer’s suggestion. We have **added new component-level ablation experiments** in the revised version, **covering prompt engineering, the verifier, the fully connected coordinator, and the forced switching mechanism**, to clarify the contributions of different modules to modeling quality, feasibility, and reasoning efficiency (Sec. 6.5, Fig. 5). The results show that:
>
> 1. Prompt engineering enables correct information extraction and modeling guidance, and its removal leads to a significant drop in both numerical-solution rate and feasibility rate;
> 2. The verifier is critical for feasibility, preventing data inconsistencies and missing constraints; otherwise, the typical infeasible errors reported in Sec. 3 reappear;
> 3. The fully connected coordinator is not a simple scheduling variant, but a necessary mechanism for routing errors to the appropriate agent; when replaced with a fixed sequence, both Num. and Feas. degrade simultaneously;
> 4. Forced switching primarily acts as a fallback to avoid repetitive debugging; it has little effect on final solution quality but reduces unnecessary retries and improves computational efficiency.
>
> In addition, the Pyomo scaffold and solver heuristics mentioned by the reviewer are low-level solver configurations that affect runtime but do not change modeling logic or feasibility determination. Therefore, our ablation focuses on components that directly impact the generation and validation of feasible solutions.

---

> > ### Author Response · Authors · 2025-11-23
> > **Response (2/2)**
> >
> > >**W3: "DS-Agent and MM-Agent are adapted from different task regimes; the paper should invest in stronger engineering-aware baselines, e.g., a single-agent Pyomo specialist with iterative verifier feedback; a graph-of-thought/skill-router agent; or a pipeline that adds identical Pyomo validation and solver heuristics for but with fixed routing, to ensure the comparison isolates coordination rather than tooling familiarity."**
> >
> > A3: Thank you for the suggestion. We agree that comparisons should avoid biases introduced by different task origins or tool familiarity. To address this, our original submission **already includes** an engineering-aware baseline. Specifically, EngiAgent (Fixed) corresponds exactly to the reviewer’s suggestion: a pipeline that adds identical Pyomo validation and solver heuristics for but with fixed routing. This design isolates the contribution of coordination without changing tool usage. As shown in Sec. 6.2, when these tool and validation components are held constant, merely altering the coordination strategy leads to substantial differences in feasibility, demonstrating that coordination itself effectively improves engineering problem-solving performance.
> >
> > We also welcome additional baseline suggestions from the reviewer, and would be glad to evaluate them for a more comprehensive comparison.
> >
> >
> > ## Questions
> >
> >
> > >**Q1: "The Modeler is Pyomo-centric. It is unclear whether the approach transfers to other modeling stacks (e.g., OR-Tools, JuMP, CasADi) or non-optimization engineering tasks (simulation, PDE-constrained design). A cross-tool study would support claims of general applicability."**
> >
> > A1: Thank you for the suggestion. Although Pyomo is used in many benchmark tasks to ensure a unified feasibility verification interface, the approach is not restricted to Pyomo-based optimization. In the revised version, we **added two non-Pyomo tasks**, demonstrating that EngiAgent can also operate beyond traditional optimization pipelines:
> >
> > - P7: **ML regression** with MILP feature search, with Pyomo used only as a solver wrapper and not as a modeling dependency.
> > - P40: Pure **ML OLS forecasting** with no optimization and **no modeling stack required**.
> >
> > EngiAgent successfully handles both tasks, showing that the coordinator does not depend on a specific modeling stack and can generalize across different modeling paradigms. These extensions only require minor adjustments to Analyzer/Modeler prompting and simple changes in solver configuration and result extraction, without modifying the core architecture. The results for these tasks are provided in Appendix G.
> >
> >
> > >**Q2: "The paper mentions expert review in dataset construction, but the feasibility labeling pipeline (human vs. automated), inter-rater agreement, and adjudication for borderline cases are under-specified. Given the centrality of feasibility, the community needs labeling guidelines, example adjudications, and an error analysis of false-feasible/false-infeasible calls."**
> >
> > A2: Thank you for the comments. In the revised manuscript, we have added Appendix B.6 and B.7 to address these concerns. In Appendix B.6, we provide a complete description of our feasibility annotation standards, including a fully human-driven dual-annotator labeling process with final adjudication by a third senior expert. We emphasize that all final labeling decisions were made exclusively by humans, ensuring no reliance on automated or LLM-based judgments, while a unified reference prompt was provided to serve as a reference. In Appendix B.7, we further include real borderline cases and adjudication analyses to illustrate how false-feasible and false-infeasible risks are identified and resolved, ensuring interpretability, objectivity, and traceability of the final labels. We believe these additions improve transparency and reproducibility.

---

> > > ### Comment · Reviewer_PFH9 · 2025-11-27
> > > **ack**
> > >
> > > Thanks for the clarification, and I keep a positive score.

---

### Official Review · Reviewer_9mVc · 2025-11-01

**Soundness:** 3
**Presentation:** 3
**Contribution:** 3
**Rating:** 6
**Confidence:** 3

**Summary:**

This paper introduces EngiAgent, a multi-agent framework designed to solve open-ended engineering problems by emphasizing feasibility. EngiAgent organizes five specialized agents (Analyzer, Modeler, Verifier, Solver, and Evaluator) under a fully connected coordinator, which enables dynamic feedback routing and flexible error correction. The study builds a benchmark dataset including 53 tasks across four engineering domains. It demonstrates substantial improvements in feasibility (up to 75.47%) compared with existing multi-agent and LLM-based baselines. Moreover, it shows that EngiAgent achieves strong cost-efficiency while surpassing previous frameworks in generating valid engineering solutions.

**Strengths:**

1. The paper introduces feasibility as the primary success criterion in engineering problem solving, which is a key but underexplored dimension in LLM-based automation.
2. The architecture is well-structured and systematically explained, with clear functional decomposition and coordination logic. Experiments are comprehensive, including ablations, cost analysis, and qualitative examples.
3. The definitions of error types (Figure 2) and evaluation dimensions are intuitive and informative.
4. The appendices provide detailed prompt designs and dataset descriptions, enhancing reproducibility.
5. The paper addresses a crucial gap between theoretical reasoning and real-world applicability of LLMs in engineering contexts.

**Weaknesses:**

1. While EngiAgent’s performance improvements are substantial, it is unclear how much of the gain stems from the coordinator architecture versus more extensive prompt engineering. This also raises questions about generalization to new tasks or LLMs without re-tuning.
2. The dataset size (53 problems) is relatively small for benchmarking general-purpose frameworks.
3. The definition of feasibility (binary feasible/infeasible) might oversimplify nuanced engineering contexts where partial or probabilistic feasibility (e.g., constraint violation tolerance, stochastic feasibility) is relevant.
4. Engineering workflows typically include iterative expert verification. EngiAgent fully automates this, but there’s no mechanism for human-guided correction or inspection, which may limit real-world deployment.
5. The system relies mainly on Pyomo-based modeling and standard solvers (GLPK, CBC, Gurobi, CPLEX). Many engineering problems involve differential equations, stochastic simulation, or finite element analysis.

**Questions:**

1. It would be valuable to report the performance of the architecture alone, without the additional prompt engineering, to better isolate its contribution. How robust is EngiAgent to different prompt wordings, model versions, or unseen problem types?
2. In many real-world engineering problems, strict feasibility may not be attainable without modifying constraints or relaxing requirements, which warrants further clarification. Can the authors give more formal definition of Feasibility?
3. How do the authors envision extending the 53-task dataset to a larger or more diverse benchmark (e.g., EngiBench)? Is there any plan to include continuous real-world or industrial data?
4. Is there any mechanism for human-in-the-loop verification or correction in EngiAgent’s workflow? If not, how do the authors prevent the system from propagating small but critical reasoning errors?
5. Does the coordinator produce interpretable reasoning traces for debugging, or is its routing behavior opaque to users?

---

> ### Author Response · Authors · 2025-11-23
> **Response (1/4)**
>
> Dear Reviewer 9mVc,
>
> Thank you for the reviewer’s positive recognition of our contributions. Our detailed responses are provided below.
>
> ## Weaknesses
>
> >**W1: "While EngiAgent’s performance improvements are substantial, it is unclear how much of the gain stems from the coordinator architecture versus more extensive prompt engineering. This also raises questions about generalization to new tasks or LLMs without re-tuning."**
>
> A1: Thank you for the reviewer's question. To distinguish performance gains from the coordinator architecture versus prompt design, we added **component-level ablation experiments** in the revised version (Sec. 6.5, Fig. 5). The ablation results show that **both prompt engineering and the fully connected coordinator significantly affect the ability to produce feasible solutions**, but in different ways. Prompt engineering helps form valid modeling components, while the coordinator enables feasibility recovery by dynamically routing errors.
>
> Regarding generalization, our method **does not require re-tuning for new tasks or different LLMs**. As shown in Table 1, using the exact same architecture and prompts (without any modification), EngiAgent achieves consistent improvements across GPT-4o, Gemini-2.5 Flash, and DeepSeek-V3-671B. This demonstrates that the gains come from the coordination and feasibility mechanisms themselves, rather than model- or task-specific customization.
>
>
> >**W2: "The dataset size (53 problems) is relatively small for benchmarking general-purpose frameworks."**
>
> A2: Thank you for the suggestion. Our current set of 53 tasks is already of very high quality and sufficiently rich to evaluate engineering feasibility in a realistic and discriminative way. Each task is built from real engineering problems sourced from top-tier journals and conferences, and includes explicit feasibility criteria, realistic data context, physical and operational constraints, and safety limits. The tasks cover four major domains of engineering practice: market and multi-agent decision-making, scheduling and resource allocation, planning and design, and control and autonomous system modeling. They further include four complementary evaluation dimensions that are essential for engineering reasoning information extraction, domain reasoning, multi-objective decision-making, and uncertainty handling. More than twenty domain experts contributed to this construction process, making the benchmark significantly more demanding to build than typical NLP datasets and, to our knowledge, the only open benchmark that evaluates engineering feasibility under real constraints.
>
> While these 53 tasks already reveal clear performance differences among models and effectively assess engineering capability, they represent only the first release of our benchmark. We have **already completed additional evaluations on extended EngiBench tasks (see Questions Q3)**, and we are actively expanding the task set further.
>
> >**W3: "The definition of feasibility (binary feasible/infeasible) might oversimplify nuanced engineering contexts where partial or probabilistic feasibility (e.g., constraint violation tolerance, stochastic feasibility) is relevant."**
>
> A3: Thank you for the comment. In this work, feasibility is used **to assess whether the most fundamental and non-negotiable engineering constraints are satisfied**. These constraints (e.g., power balance, safety limits, stability ranges) define whether a solution can be executed at all; if any of them is violated, the solution is not deployable in practice, regardless of its proximity to feasibility. Therefore, our binary criterion targets this minimum requirement, distinguishing between implementable and non-implementable solutions. To ensure transparency, we have added **Appendix B.6–B.7** in the revision, which documents the annotation protocol and real borderline cases.

---

> > ### Author Response · Authors · 2025-11-23
> > **Response (2/4)**
> >
> > >**W4: "Engineering workflows typically include iterative expert verification. EngiAgent fully automates this, but there’s no mechanism for human-guided correction or inspection, which may limit real-world deployment."**
> >
> > A4: Thank you for raising this important point. In real engineering workflows, multiple levels of automation typically coexist, from human-in-the-loop processes to partially and fully automated pipelines. Our work examines whether LLM agents can achieve end-to-end automation for a well-defined subset of engineering tasks. This does not imply that practical deployment would eliminate human oversight. Instead, human-in-the-loop, human–agent collaboration, safety inspection, and error correction remain crucial topics for future research.
> >
> > Our aim in this paper is to evaluate the feasibility and potential of fully automated reasoning within controlled engineering settings, and to understand how far an agentic workflow can operate without human intervention. As shown in Table 1, EngiAgent’s feasibility rate and numerical rate are strongly aligned, indicating that end-to-end automation is achievable for many structured tasks.
> >
> > This exploration provides practical value in scenarios where scalability, repeatability, and minimal overhead are important, while still leaving substantial room for future work. Broader issues such as human–agent collaboration, structured expert oversight, adaptive correction mechanisms, safety assurance, and real-world deployment protocols are indeed critical directions for engineering AI systems. These topics, however, sit at a different layer of the automation stack and are beyond the scope of the this paper.
> >
> > >**W5: "The system relies mainly on Pyomo-based modeling and standard solvers (GLPK, CBC, Gurobi, CPLEX). Many engineering problems involve differential equations, stochastic simulation, or finite element analysis."**
> >
> > A5: Thank you for the suggestion. **Pyomo is used merely as a unified feasibility and solver interface, rather than a requirement of the method**. EngiAgent’s core mechanism is feasibility-driven coordination and error routing, which remains **decoupled from any specific modeling or solver stack**.
> >
> > To further verify that the method does not rely on Pyomo and standard solvers, we added two tasks in the revision:
> > - P7 (ML-driven model selection): A MAPE-based regression task where the model is learned via ML estimation, and a MILP is used only as a feature-selection search tool; Pyomo is used merely as a solver interface, not for symbolic modeling.
> > - P40 (Pure ML forecasting): An OLS regression task solved entirely using numerical ML methods without any Pyomo usage or optimization backend.
> >
> > Results are provided in Appendix G, and these tasks were enabled with only minor adjustments to Analyzer/Modeler prompting and solver configuration, without modifying the core architecture. This demonstrates that EngiAgent is not tied to Pyomo-based modeling and can naturally extend to more diverse engineering domains in future work.
> >
> > ## Questions
> >
> > >**Q1: "It would be valuable to report the performance of the architecture alone, without the additional prompt engineering, to better isolate its contribution. How robust is EngiAgent to different prompt wordings, model versions, or unseen problem types?"**
> >
> > A1: Thank you for the reviewer’s suggestion. We **report the architecture without full prompt engineering through component-level ablations in the revised version (Sec. 6.5, Fig. 5)**. The results show that keeping the coordinator but removing prompt engineering causes a large drop in feasibility, meaning that the architecture alone cannot compensate for missing or unstructured modeling inputs. Prompt engineering and coordination are complementary: prompts specify the modeling structure, while the coordinator restores feasibility through error routing.
> >
> > **Regarding robustness to different prompt wordings, model versions, or unseen problem types, EngiAgent does not require re-tuning or rewriting of prompts**. As shown in Table 1, the exact same prompt and architecture (no modifications) work across GPT-4o, Gemini-2.5 Flash, and DeepSeek-V3-671B, achieving consistent improvements. The method does not rely on training data or handcrafted templates tied to specific tasks, and therefore generalizes without prompt rewriting or model-specific tuning.

---

> > > ### Author Response · Authors · 2025-11-23
> > > **Response (3/4)**
> > >
> > > >**Q2: "In many real-world engineering problems, strict feasibility may not be attainable without modifying constraints or relaxing requirements, which warrants further clarification. Can the authors give more formal definition of Feasibility?"**
> > >
> > > A2: Thank you for the question. In this work, **a solution is considered feasible only if it satisfies all mandatory engineering constraints**. As stated in Section 3: *A solution that is logically consistent and mathematically valid becomes meaningless if it violates data consistency, physical laws, operational safety, or procedural constraints*.
> > >
> > > To clarify this more explicitly, we have **added a formal statement** in the revision, consistent with the principles already stated in Section 3: *In this work, a solution is considered feasible only if it satisfies all mandatory engineering constraints required for execution in real systems, such as data consistency, physical requirements, safety limits, and procedural rules.* In addition, relaxing or modifying constraints to obtain “near-feasible” solutions effectively changes the original problem, rather than solving it.
> > >
> > > For transparency and reproducibility, we have additionally included **Appendix B.6–B.7**, which provide the feasibility annotation procedure and real borderline examples.
> > >
> > >
> > > >**Q3: "How do the authors envision extending the 53-task dataset to a larger or more diverse benchmark (e.g., EngiBench)? Is there any plan to include continuous real-world or industrial data?"**
> > >
> > > A3: Thank you for the reviewer’s suggestion. In the revised version, we have **added tasks from EngiBench for evaluation**. Since the core goal of EngiAgent is to produce feasible numerical solutions, we filtered the EngiBench tasks by keeping constraint-based modeling that require feasible solutions, and removing purely analytical, unconstrained modeling, or tasks without a definable feasibility criterion. In total, 19 tasks are suitable for evaluating EngiAgent. Based on EngiAgent’s dataset construction process, we added explicit feasibility constraints and verification criteria for each task. All results below are obtained using EngiAgent on GPT-4o. For a clearer comparison, we also report EngiAgent’s results on our original dataset as a reference baseline.
> > >
> > > **Table: Numerical and Feasibility Rates**
> > >
> > > | Dataset            | Num. ↑ | Feas. ↑ |
> > > |--------------------|--------|---------|
> > > | Original Dataset   | 66.04% | 64.15%  |
> > > | EngiBench          | 63.16% | 52.63%  |
> > >
> > > **Table: Total Performance**
> > >
> > > | Dataset            | IE ↑ | DR ↑ | MO ↑ | UH ↑ | Avg. ↑ |
> > > |--------------------|------|------|------|------|--------|
> > > | Original Dataset   | 8.67 | 7.74 | 7.05 | 7.41 | 7.72   |
> > > | EngiBench          | 7.78 | 6.36 | 6.14 | 6.20 | 6.62   |
> > >
> > > **Table: Feasible-Only Performance**
> > >
> > > | Dataset            | IE ↑ | DR ↑ | MO ↑ | UH ↑ | Avg. ↑ |
> > > |--------------------|------|------|------|------|--------|
> > > | Original Dataset   | 8.86 | 8.13 | 7.53 | 7.94 | 8.12   |
> > > | EngiBench          | 7.50 | 6.72 | 7.96 | 7.50 | 7.42   |
> > >
> > >
> > > In addition, these **evaluation tasks originate from real engineering and industrial research** published in top-tier journals and conferences, rather than being artificially constructed examples. Therefore, our experiments are already conducted on real-world / industrial data and task settings, and future extensions will continue to include more constrained and verifiable open-ended engineering tasks.
> > >
> > > >**Q4: "Is there any mechanism for human-in-the-loop verification or correction in EngiAgent’s workflow? If not, how do the authors prevent the system from propagating small but critical reasoning errors?"**
> > >
> > > A4: Thanks for the question. Our workflow does not involve any human-in-the-loop, which is in fact one of EngiAgent’s strengths. The system uses the Verifier, Evaluator, Fully Connected Coordinator, and Memory module to form a multi-layer automated checking mechanism that replaces manual supervision.
> > >
> > > The Verifier ensures semantic and constraint consistency before solving; the Coordinator routes errors to the correct agents and **prevents error propagation**; and the Memory module records errors and fix strategies to avoid repeated mistakes. Together, these components provide human-like oversight in a fully automated manner.
> > >
> > > Our results in Table I confirm this: across all three models, the numerical-solution rate and feasible-solution rate are almost identical. This shows that EngiAgent can maintain feasibility reliably without human intervention.

---

> > > > ### Author Response · Authors · 2025-11-23
> > > > **Response (4/4)**
> > > >
> > > > >**Q5: "Does the coordinator produce interpretable reasoning traces for debugging, or is its routing behavior opaque to users?"**
> > > >
> > > > A5: Thank you for the question. The coordinator’s **routing behavior is interpretable** rather than opaque. For each routing action, the system records structured natural-language reasoning traces, including the triggering signal, supporting evidence, and the agent-selection rationale, and stores the corresponding versioned intermediate inputs and outputs in the Memory module to enable debugging-oriented traceability and post-hoc inspection. To further clarify this, representative examples have been added in the **Appendix E "Coordinator Routing Trace Examples"**.

---

### Author Response · Authors · 2025-12-01
**General Response to the AC and Reviewers (RBoR, PFH9, 9mVc)**

Dear Area Chair and Reviewers,

We sincerely thank the AC and the reviewers for their valuable time and constructive feedback. Three reviewers (RBoR, PFH9, 9mVc) consistently provided positive evaluations from the initial review through the discussion phase and up to the end of the rebuttal. We would like to take this opportunity to summarize (1) the key points highlighted in the reviews and (2) the updates included in the rebuttal revision.

>**Key points consistently recognized in the reviews:**

1. The paper introduces **feasibility as the primary criterion** for engineering-oriented LLM problem solving, highlighting a **critical and underexplored gap** between theoretical reasoning and real-world applicability and providing a structured summary of **common engineering failure patterns**.

2. The EngiAgent architecture is well structured and systematically explained, with a fully connected coordinator that analyzes feedback, routes errors to the appropriate role-specific agent, and guides the solution toward feasibility.

3. Across models and task domains, the method **achieves significant and stable absolute improvements** in feasibility over strong baseline agents. The evaluations are comprehensive, including ablations, cost analysis, and qualitative workflow examples.

4. The paper shows that text-based LLM-as-a-judge scoring often diverges from true engineering feasibility and explains why textual plausibility does not guarantee implementable solutions.

>**The rebuttal revision includes the following updates:**

1. Added component-level ablation experiments (Section 6.5), evaluating the independent contributions of the coordinator, verifier, forced switching, and prompt engineering.

2. Released the complete feasibility annotation procedure (Appendix B.5–B.7), including dual-annotation, expert adjudication, and borderline case documentation.

3. Added interpretable routing examples for the coordinator (Appendix E), illustrating routing triggers, decision signals, and underlying reasoning.

4. Expanded baseline configuration descriptions (Appendix F), specifying consistent solver access, execution budgets, and runtime settings across methods.

5. Added two cases (Appendix G) to demonstrate extensibility to ML-driven and non-optimization modeling paradigms.

---

### Meta-Review · Area_Chair_3Led · 2026-01-01

**Summary:**

This submission proposes EngiAgent, a multi-agent framework for open-ended engineering problem solving where feasibility (implementability under data/physical/operational constraints) is treated as the primary success metric. The system decomposes the workflow into five roles and introduces a Fully Connected Coordinator with shared memory to dynamically route errors and feedback across roles, rather than relying on a fixed pipeline. The paper also contributes a 53-task benchmark spanning four engineering domains and reports large feasibility gains across multiple LLM backbones, alongside analysis highlighting the misalignment of text-based judging with true feasibility and a cost/efficiency study.

The review set consists of three marginally positive reviews (all scored 6) and one strong reject (score 0). Overall, the work targets a practically important gap, that is, LLMs often produce plausible but infeasible "solutions" for engineering tasks. And it presents a clearly described system with strong headline improvements.

The authors' rebuttal adds useful ablations, expanded annotation protocol details, and additional discussion of routing traces and baseline fairness, which address several reviewer questions to a reasonable extent. However, There are concerns that collectively limit my confidence in the strength and generality of the method: 1) feasibility evaluation rigor and reproducibility is concerning, 2) verifier semantics and risk of specification drift is unclear, and 3) method justification remains largely empirical. Thus, I recommend rejection.

**Reviewer Concerns:**

The concerns from reviewers with positive scores are addressed, as mentioned by the reviewers, though I think it is not fully addressed. The concerns from the reviewer with score 0 are not addressed well. They are 1) role decomposition not principled; 2) coordinator is heuristic with no guarantees; 3) verifier is "not verification".

**Reviewer Scores:**

The reviewers with positive scores keep their positive ones. The reviewer with score 0 maintains his/her stance.

---

### Decision · Program_Chairs · 2026-01-26

Reject